# Worm orthologues of cytokinesis-associated proteins CIT and ASPM regulate neuronal microtubule dynamics and polarity in *C. elegans*

Sunanda Sharma[1]*, Keerthana Ponniah[1]¤, Ishanee Bandyopadhyay[1],
Devyani Vadawale[2], Sandhya P. Koushika[2], Anindya Ghosh-Roy[1]*

**1** Department of Cellular & Molecular Neuroscience, BRIC-National Brain Research Centre, Manesar, Haryana, India, **2** Department of Biological Sciences, Tata Institute of Fundamental Research, Mumbai, India

¤ Current address: Clem Jones Centre for Ageing Dementia Research, Queensland Brain Institute, The University of Queensland, Brisbane, Australia
* anindya@nbrc.ac.in (AG-R); sunandasharma971@gmail.com (SS)

## Abstract

The polarized architecture of neurons is intricately associated with the modulation of microtubule dynamics. Over the years, several microtubule-associated factors that regulate neuronal polarity have been identified. However, the precise details of how microtubule arrangement and stability are established in axons and dendrites are not clearly understood. To uncover the relevant factors involved in the biological pathways governing microtubule regulation in neurons, we conducted a suppressor screen using the neuronal ectopic extension phenotype caused by the loss of the kinesin-13 family microtubule depolymerizing protein KLP-7 in *C. elegans*. Interestingly, apart from eleven variants of α (*mec-12*) and β (*mec-7*) tubulins, we isolated a variant of cytokinesis-associated protein, *W02B8.2*/*citk-1,* the suggested kinase-less orthologue of mammalian citron-rho interacting kinase (CIT). Little is known about the role of CIT in microtubule regulation in post-mitotic neurons. In this study, we found that the kinase-less worm orthologues of CIT, *citk-1* and *citk-2,* redundantly modulate microtubule stability in the axon-like anterior process and maintain the population of plus-end-out microtubules in the dendrite-like posterior process of the PLM mechanosensory neurons in a cell-autonomous manner. In the absence of *citk-1* and *citk-2*, PLM neurons exhibit variable morphological defects, including neurite growth and synaptic branch defects. Moreover, we find that CITK-1/2 work in the same genetic pathway as ASPM-1 (the worm homolog of mammalian ASPM (abnormal spindle-like microcephaly-associated protein)) to modulate plus-end dynamics of microtubules in PLM neurons. Our findings suggest that the cytokinesis-associated CITK-1/2 and ASPM-1 have non-mitotic roles in regulation of microtubules in differentiated PLM neurons.

**Data availability statement:** All relevant data are in the manuscript and its supporting information files.

**Funding:** This work is supported by the NBRC core fund from the Department of Biotechnology, the DBT/Wellcome Trust India Alliance Senior Fellowship (Grant # IA/S/22/1/506243 to AGR), and DAE grant (Grant# 1303/2/2019/R&D-II/DAE/2079 to SPK). The funders had no role in study design, data collection and analysis, decision to publish, or preparation of the manuscript.

**Competing interests:** The authors have declared that no competing interests exist.

## Author summary

Neurons are the fundamental unit of the nervous system. Their complex structure and function rely on a cell skeleton, an internal framework of long, rod-like structures called microtubules. Microtubules can grow and shrink by the addition or removal of their building blocks, tubulin heterodimers. Disruptions in this structure are linked to neurodevelopmental and neurodegenerative disorders. Using touch neurons of *C. elegans,* we identified that the proteins CITK-1/2 and ASPM-1, which are the worm counterparts of mammalian microcephaly-associated proteins CIT and ASPM, regulate neuronal microtubules. We found that CITK-1 and CITK-2 bear structural homology with the neuronal isoform of mammalian CIT (CIT-N) and lack an enzymatic domain. Using live imaging and genetic manipulations, we found that CITK-1 and ASPM-1 trail the growing plus-ends of microtubules to regulate plus-end dynamics. Loss of these proteins causes various morphological defects in neurons. Intriguingly, CIT and ASPM-1 are traditionally known for their roles in cell division. Our findings identify how microcephaly- and cell-division-associated proteins are repurposed to modulate neuronal cytoskeleton.

## Introduction

The fundamental unit of the nervous system, a neuron, is a highly polarized cell. Its ability to transmit information unidirectionally over long distances in neural networks relies on its structural and functional compartmentalization into dendrites and axons, the information receiving and transmitting processes [1]. One of the most sought after questions in neurobiology is how this elaborate architecture is developed and maintained throughout the lifespan of an organism. Previous studies have demonstrated that the regulation of neuronal polarity is closely linked to the arrangement of microtubule polarity and dynamics within neuronal compartments [2–6]. Microtubules are tubular, hollow, heteropolymers of α-β tubulin dimers, arranged in a head-to-tail manner, rendering them an intrinsic polarity. [7,8]. The visibly dynamic end with a β-tubulin exposed is called the plus end, and the less dynamic end with an α-tubulin exposed is called the minus end [9]. In the vertebrate neuronal system, *in vitro* studies have shown that axons have a unipolar plus-end-out (plus-end-distal to cell body) microtubule arrangement, while dendrites have a mixed microtubule polarity [10,11]. *In vivo* studies in invertebrate neurons also show a plus-end out microtubule arrangement in axons. However, the dendritic compartment in invertebrates mostly shows minus-end out microtubule polarity, and sometimes mixed [2,12–14]. Although little is known about how this compartmentalized microtubule arrangement and dynamics are achieved in neurons, recent studies have found novel insights into the roles of microtubule retrograde flow and polarity reversal in defining dendritic microtubule polarity [15–17].

PLOS Genetics

Over the years, several microtubule-associated factors, including microtubule nucleation factors (such as g-TURC), polymerization factors (for instance, tubulins, CRMP-2/UNC-33), microtubule anchoring proteins (TRIM46, UNC-44/Ankyrin-G), and regulators of microtubule-end dynamics (plus-end-binding proteins EBP-2/3, minus-end binding protein Patronin/CAMSAP1/2/3), have been identified as regulators of neuronal microtubule dynamics and arrangement [6,18–22]. The microtubule cross-linking and anchoring factors, such as TRIM46 and Ankyrin-G, organize the microtubules in parallel bundles with plus-end-out polarity in the axon initial segment [23–26]. The microtubule end-binding proteins function as microtubule-stabilizing factors in axons and dendrites [21,22,27,28]. While Kinesin-13 family microtubule depolymerizing factor KIF2A/KLP-7 is a gatekeeper of dendritic identity in mouse, fish, and *C. elegans* neurons [2,4,5,29,30,31]. In the absence of kinesin-13, microtubules are hyper-stabilized, leading to reorganization of dendritic microtubules in a plus-end-out orientation [2,14]. However, it's not completely clear how the microtubules in the distal axon, which are farther away from the AIS (axon initial segment), remain bundled and optimally stabilized. Furthermore, our understanding of how microtubules retain a mixed polarity organization within dendritic compartments is fragmentary.

The mechanosensory neurons of *C. elegans* are an excellent model to study the regulation of microtubule dynamics and polarity in neurons [32–34]. Several microtubule-regulating factors, as well as molecules regulating post-translational modifications of tubulins, have been identified and studied in the ALM and PLM mechanosensory neurons [35–42]. The PLM (posterior lateral microtubule) neuron is a well-characterized bipolar neuron that, under the guidance of Wnt signaling gradient, polarizes and extends its neurites in the antero-posterior axis [2,43–46]. Its anterior neurite exhibits plus-end out microtubule arrangement, similar to vertebrate axon, while the posterior short neurite displays a mixed microtubule polarity, similar to vertebrate dendrites [2]. Any perturbation in the microtubule dynamics lead to distinct morphological changes in these neurons [2,35]. For instance, loss of Kinesin-13 family microtubule depolymerizing factor KLP-7 results in hyper-stabilization of microtubules and hence, ectopic neurites sprouting from mechanosensory neurons of mutant animals, including overgrowth of PLM posterior process [2]. The ALM (anterior lateral microtubule) neurons, on the other hand, are largely unipolar; however, in *klp-7(0)* animals, ALM neurons develop a visibly striking posterior ectopic extension [2,29]. These ectopic growths can be suppressed by destabilizing microtubules [2,35]. We hypothesized that the *klp-7(0)* mutant phenotype, combined with the powerful genetic toolbox of *C. elegans* [47], could be a promising platform for designing a suppressor screen aimed at identifying key regulators of neuronal microtubule dynamics, which could enhance our understanding of how microtubule dynamics and polarity are regulated in neurons.

In this study, we isolated twenty-six suppressor variants in a screen and mapped twelve of these to the loci of mechanosensory neuron-specific tubulins *mec-12* (α-tubulin) and *mec-7* (β-tubulin). Incidentally, we isolated a variant of *W02B8.2*/*citk-1,* the proposed kinase-less worm orthologue of mammalian citron-rho kinase-associated protein/CIT/MCPH17 (autosomal recessive primary microcephaly-17), that is typically associated with cytokinesis. Variants of CIT/MCPH-17 are also associated with MCPH (Primary autosomal recessive microcephaly), a neurodevelopmental disorder [48–50].

Mammalian *Cit* gene majorly encodes two isoforms: a long isoform CIT-K, which has an N-terminal kinase domain and is ubiquitously expressed in proliferating cells, and a short isoform, CIT-N, which lacks the N-terminal kinase domain and is expressed exclusively in the post-mitotic and differentiated cells of the central nervous system [51,52].

It has been shown that the longer isoform CITK orchestrates midbody maturation and organization of spindle microtubule arrays in dividing cells [53–56]. Furthermore, in-situ hybridization experiments, showed CITK expression in a specific neuronal population in the cortex and thalamus, suggesting a role in neuronal differentiation [57]. The neuronal cell lines studies suggest the involvement of CITK as well as CITN, the kinase-less isoform of CIT, in limiting neurite outgrowth [57–59]. Additionally, *in-vitro* cultures and *in vivo* knockdown studies show that CIT-N regulates dendritic spine density and maturation. Notably, CIT-N is shown to be localized with spine-associated Golgi compartments and post-synaptic densities *in vitro* [57,59–61]. However, the roles of CIT proteins in the regulation of microtubule dynamics in post-mitotic neurons have not been explored.

The suggested worm ortholog lacks the N-terminal kinase domain and shares a closer structural similarity with the CIT-N isoform. Here, using live imaging of EBP-2::GFP reporter we show that *citk-1,* along with its paralog *citk-2,* limits the microtubule dynamics in the axon-like anterior process of PLM mechanosensory neurons. It also maintains the plus-end-out microtubule population in the dendrite-like posterior process in a cell-autonomous manner. Furthermore, loss of both citron kinase proteins results in morphological defects in these neurons. Additionally, cytokinesis-associated protein ASPM-1, an orthologue of mammalian ASPM (Abnormal Spindle-like Microcephaly-Associated Protein, MCPH5), is a known partner of CIT in microtubule regulation during cell division. We found that *aspm-1(ts)* mutant phenocopies the *citk* microtubule defects. Both CITK-1 and ASPM-1 co-localize with the plus-tip protein EBP-2, and act in the same pathway to maintain the growing plus tips in the PLM neurite. These findings suggest that cytokinesis-associated proteins *citk-1/citk-2* and *aspm-1* have a post-differentiation role in the regulation of microtubule dynamics in neurons.

## Results

### A modifier screen using the ectopic extension phenotype of *klp-7(0)* identifies neuronal microtubule regulators, including tubulin variants

Previous studies have shown that loss of KLP-7, a kinesin-13 family microtubule depolymerizing protein, leads to ectopic neurite extensions in mechanosensory neurons of *C. elegans* [2,29]. Most notably, the ALM mechanosensory neurons exhibit a striking posterior extension (Fig 1A and 1B'). In wild-type animals, the two bilateral ALM neurons are typically unipolar with a long anterior process; any posterior process is usually shorter than twice the length of the ALM cell body (Fig 1A and 1B') [32]. However, in *klp-7(0)* mutant animals, this posterior process overgrows and may even extend beyond the cell body of the posteriorly situated PVM mechanosensory neuron (Fig 1A and 1B') [2,29]. Furthermore, this phenotype can be suppressed by destabilization of microtubules, either by colchicine treatment or by a second-site mutation in mechanosensory neuron-specific tubulins, *mec-7*(β-tubulin), or *mec-12* (α-tubulin) [2,14,35].

This suggests that the *klp-7(0)* ectopic extension can serve as a platform to isolate novel microtubule regulators, enhancing our understanding of neuronal microtubule regulation (Fig 1A). Hence, we conducted an EMS mutagenesis screen and isolated twenty-six unique suppressor variants from 12,422 F1s (Fig 1A–1C). We classified the ALM ectopic extension into three categories based on their length: No extension (WT-like posterior extension), Mild extension (extension does not reach the vulva), or Strong extension (extension reaches or crosses the vulva) (Fig 1B–1B'). We then categorized the isolated suppressor variant as either a complete suppressor (no strong extensions in the population, e.g., *shr10, shr15*) or a mild suppressor (e.g., *shr11, shr1)* (Fig 1C).

We used EMS density deep sequencing mapping to identify the molecular nature of these variants (please see the methods section for details). We mapped eleven variants to mechanosensory neuron-specific tubulins. The *shr15, shr20,* and *shr25* suppressors mapped to *mec-7* (β-tubulin) (Figs 1D, S1A and S1C), and *shr1, shr9, shr10, shr14, shr17, shr19, shr21,* and *shr22* mapped to *mec-12* (α-tubulin) (S1B and S1D Fig). We validated these results by expressing a wild-type copy of the gene of interest (*mec-7* in this case) in the suppressor backgrounds, which successfully restored the *klp-7(0)* phenotype (Fig 1D and 1E).

In summary, we isolated twenty-six unique suppressors of *klp-7(0)* ectopic extension phenotype. Eleven suppressor mutants mapped to mechanosensory neuron-specific tubulins *mec-7* and *mec-12.* Since the loss of *mec-7* and *mec-12* is known to destabilize microtubules [2,14,35]. This finding validates our screen as a robust method for identifying neuronal microtubule regulators. Beyond these expected factors, we also identified some interesting candidates, including an RNA-binding protein, *mbl-1* [14], and a cytokinesis-associated factor, *citk-1*, which we will be discussing in detail in the following sections.

**Fig 1. Isolation and mapping of suppressors of *klp-7(0)* ALM ectopic extension phenotype. (A)** Schematics illustrates the hypothesis of EMS genetic screening. The wild-type animals with a steady-state microtubule dynamics exhibit no posterior extension from ALM mechanosensory neuron.

Loss of function mutations in the microtubule depolymerizing protein KLP-7 (orange structure) leads to a posterior ectopic extension (drawn in purple) from the ALM cell body, associated with stable microtubules in these animals. The stable microtubules in the *klp-7(0)* mutant can be destabilized by EMS-induced second-site mutations in other microtubule regulators. The *klp-7(0)* ectopic extension phenotype would be reduced or absent in these suppressor mutant backgrounds. The blue and gray spheres represent α and β tubulins, respectively, which form the structure of microtubules. The purple ovals represent minus-end binding protein and the green hourglass like-structures represents plus-end binding proteins. The deep sky-blue hexagon depicts a probable microtubule stabilizing protein being mutated in suppressor mutant. Microtubule dynamics scheme is adapted from [62]. **(B-B')** Representative confocal images and schematics of ALM and PLM neurons in the L4 staged wild-type, *klp-7(0)* mutant, a suppressor *klp-7(0); mec-7(shr20)* and a *klp-7(0); mec-7(shr20); mec-7* (genomic DNA)[+] animals. The anterior end of animal is placed left, and vulva is positioned downwards in this and following figures. The ALM and PLM mechanosensory neurons of these animals express GFP (green fluorescent protein) under mechanosensory neuron specific promoter, *pmec-7* (*pmec-7::GFP, muIs32)*. (0) represents the loss of function deletion allele *(tm2143)* of *klp-7*. The white arrowhead labels the no ectopic extension phenotype in wild-type animal. The yellow and pink arrowheads point to strong and mild posterior extensions from ALM in *klp-7(0)* animals that ends either before or after the vulval position landmark(*), respectively. The red arrowhead points to suppression of ALM ectopic extension phenotype in *klp-7(0); mec-7(shr20)* mutant which is rescued (yellow arrowheads) in *klp-7(0); mec-7(shr20); shrEx396(mec-7(gDNA)[+])* animals. A double-headed white arrow is drawn along the length of the PLM posterior process. **(C)** Quantification of the percentage of ALM neurons with ectopic extension in the *klp-7(0)* and the suppressor backgrounds *klp-7(0); shr#*, where *shr#* is the allele number. N = 3–4 independent replicates, n (number of neurons) = 45-55. **(D)** A schematic representation of the fosmid WRM062bA08, which includes the complete genomic cassette of the *mec-7 gene*, as shown. **(E)** Quantification of the percentage of ALM neurons with ectopic extension in the wild-type, *klp-7(0)*, *klp-7(0); mec-7(shr15)*, *klp-7(0); mec-7(shr20)*, *klp-7(0); mec-7(shr25)*, *klp-7(0); mec-7(shr15); mec-7(gDNA) [+]*, *klp-7(0); mec-7(shr20); mec-7(gDNA) [+]* and *klp-7(0); mec-7(shr25); mec-7(gDNA) [+]* backgrounds, where g stands for genomic and *mec-7(gDNA)* is the fosmid WRM062bA08. N = 3–5 independent replicates, n (number of neurons) = 45-50. For C and E, ***, P < 0.001; **, P < 0.01, P values from 2x2 Fisher's exact test comparing number of animals with No extension phenotype and number of animals with a Neurite defect (mild-extension + strong extension) between compared genotypes.

## Mutations in a previously uncharacterized protein *W02B8.2/citk-1* suppress the ectopic extension phenotype of *klp-7(0)* mutant

We mapped the partial suppressor *shr11* to a proline to leucine substitution at the 855th amino acid of the W02B8.2 locus on Chromosome II (Figs 2A, 2B and S2A). *W02B8.2* is a proposed ortholog of mammalian citron-rho interacting kinase, a protein known for its role in midbody organization during cytokinesis [66].

We validated that *shr11* is an allele of *W02B8.2* through several validation experiments. First, an extrachromosomal expression of the wild-type *W02B8.2* gene using fosmid WRM062dD08 (which has a complete genomic cassette of *W02B8.2*) (Fig 2B) restored the *klp-7(0)* phenotype (Figs 2A–2A', 2D and S2B). Second, we found that a known deletion allele of *W02B8.2*, *ok2328* (S2I Fig), reproduced the partial suppression of the *klp-7(0)* phenotype (Fig 2D). Finally, using a touch-neuron-specific promoter, *pmec-4,* we demonstrated that touch neuron-specific expression of *W02B8.2* cDNA rescues the *klp-7(0)* phenotype (Figs 2A–2A', 2D and S2B). This implies a cell-autonomous role of *W02B8.2* in suppression of the *klp-7(0)* phenotype. We also validated that this transgenic expression of *W02B8.2* does not cause any morphological defects in ALM mechanosensory neurons of wild-type animals (S2K Fig). This confirmed that the observed ALM ectopic extension in suppressor mutant on transgenic expression of W02B8.2 is indeed *klp-7(0)* phenotype rescue.

The sequence and structural blast analysis suggest that W02B8.2 is a kinase-less orthologue of neuronal isoform of mammalian citron-rho-interacting kinase (CIT-N) (S2E Fig). We identified a second paralog in worms, F59A6.5, which shares 66% sequence homology with W02B8.2 (S2E and S2F Fig). Using the protein databases [63–65], we identified that W02B8.2 and F59A6.5 share structural homology with CIT-N [67] (Fig 2C). W02B8.2 and F59A6.5, like CIT-N, lack an N-terminal kinase domain but contain conserved regulatory regions. This includes an extended coiled-coil domain, a phorbol ester/DAG type Zn finger domain (C1 domain), a pleckstrin homology (PH) domain, and a Citron-Nik-1 homology (CNH) domain (Figs 2C, S2G and S2H). Based on these structural similarities, we named these loci *citk-1* and *citk-2,* respectively.

The identification of a paralog *F59A6.5/citk-2* and observed partial suppression of *klp-7(0)* phenotype by *W02B8.2/citk-1* mutations suggest a functional redundancy. Therefore, we assessed whether the *citk-2* mutation phenocopies the suppression phenotype of *citk-1(0)* in the *klp-7(0).* We used a frameshift deletion variant of *citk-2, ok2885*, to evaluate the suppression (S2J Fig). The loss of *citk-2* in the *klp-7(0)* background significantly reduced the length of ALM ectopic extension (Fig 2A–2A' and 2E), suggesting that *citk-2(ok2885)* is also a partial suppressor of *klp-7(0)* phenotype.

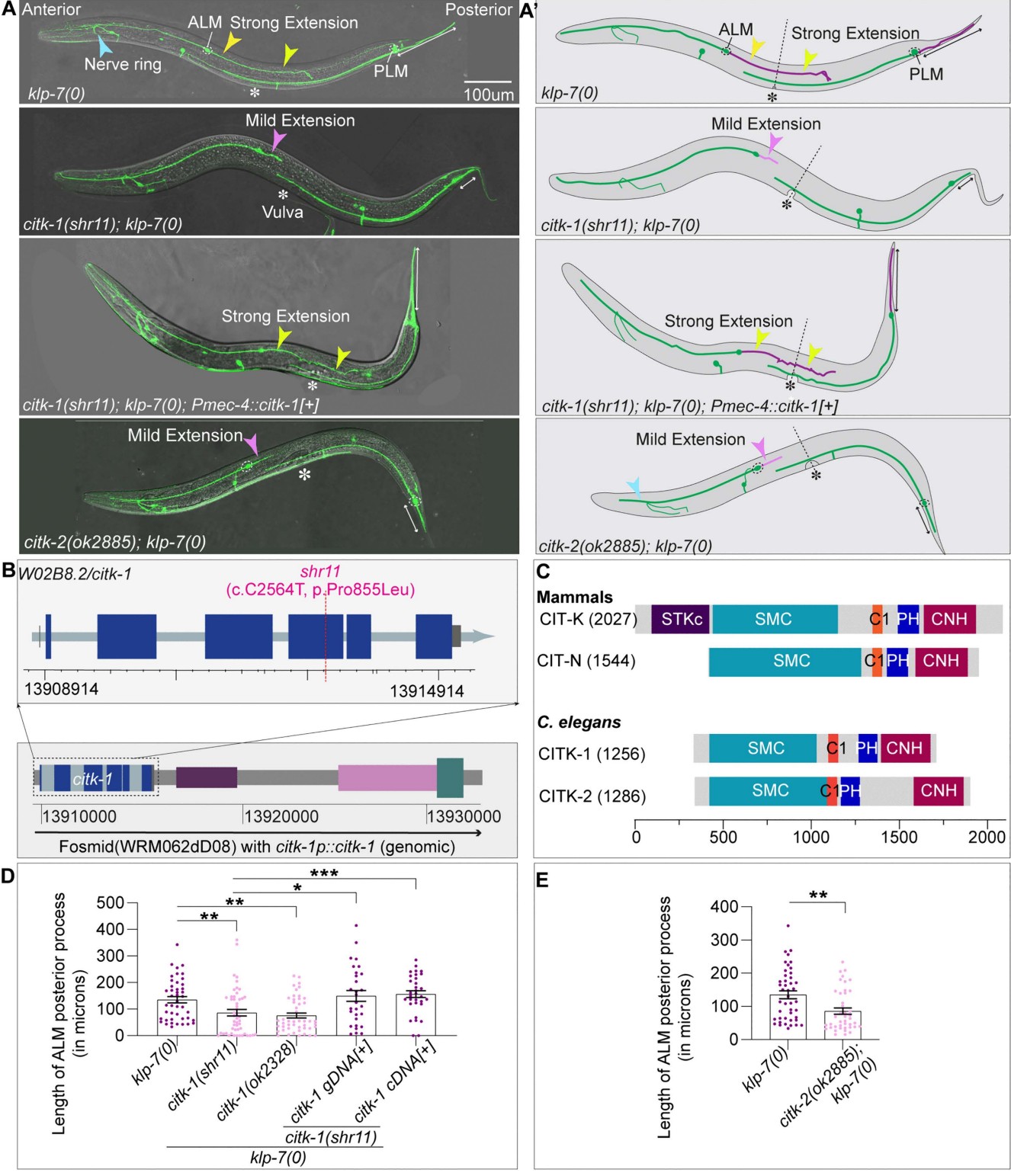

**Fig 2. Identification and characterization of CITK-1 and CITK-2 as partial suppressors of *klp-7(0)* phenotype. (A-A')** Representative confocal images of the ALM and PLM mechanosensory neurons in the L4 stage *klp-7(0), citk-1(shr11); klp-7(0), citk-1(shr11); klp-7(0); shrEx432(pmec-4::citk-1[+])* and *citk-2(ok2885); klp-7(0)* mutant animals along with their respective schematics. These neurons are labeled green by *pmec-7::GFP* reporter. The

yellow-arrowheads point to the strong ectopic extension from the ALM neuron that crosses the vulval landmark, labeled by an asterisk (*) in *klp-7(0)* animals and the *citk-1(0); klp-7(0)* animals with an extrachromosomal expression of *pmec-4::citk-1*. *pmec-4* is a mechanosensory neuron specific promoter. The pink arrowheads point to the mild ectopic extension in *citk-1(shr11); klp-7(0)* and *citk-2(ok2885); klp-7(0)*. The double-headed white arrow extends along the length of PLM posterior process. **(B)** The schematic of the exon and intron of the *W02B8.2/citk-1* gene and the genomic positions of the identified candidate mutation within the *citk-1* locus of the *shr11* suppressor. At the bottom is a representation of the fosmid WRM062dD08 which contains the complete genomic cassette of the *citk-1* gene. **(C)** The schematics of the two isoforms of mammalian citron-rho interacting kinase proteins and the *C. elegans* paralogs CITK-1 and CITK-2. Adapted from [63–65]. **(D-E)** Quantification of the absolute length of ALM ectopic extension (D) in the *klp-7(0)*, *citk-1(shr11); klp-7(0)*, *citk-1(ok2328); klp-7(0)*, *citk-1(shr11); klp-7(0); citk-1* genomic DNA [+], *citk-1*(shr11); *klp-7(0); citk-1* cDNA [+] backgrounds and (E) in *klp-7(0)* and *citk-2(ok2885); klp-7(0)* backgrounds, where genomic *citk-1* is the *citk-1* fosmid WRM062dD08 and the *citk-1(cDNA)* is expressed under promoter *pmec-4*. For D, N = 3 independent replicates, n (number of neurons) = 30-50. For E, n = 38-44, N = 3. For D-E, Error bars represent SEM (Standard error mean), ***, P < 0.001; **, P < 0.01 and *, P < 0.05, for **(D)** P values from Kruskal Wallis test followed by Dunn's multiple comparisons. For **(E)** P values from unpaired two tailed T test.

As previously reported, *klp-7(0)* mutant animals also display an overgrowth of the posterior process of PLM neurons [2] (Fig 2A–2A'). These mechanosensory neurons run ventrolaterally in the posterior half of the worm. Unlike ALM neurons, PLM neurons are bipolar with a long axon-like anterior process and a short dendrite-like posterior process extending into the tail of the animal (Fig 2A–2A'). We observed that mutations in either *citk-1* or *citk-2* in the *klp-7(0)* background can significantly suppress the overgrowth of the PLM posterior process (Figs 2A–2A', S2C and S2D). Furthermore, the PLM overgrowth was rescued by an extrachromosomal expression of either a genomic or cDNA copy of wild-type *citk-1* in the suppressor background (Figs 2A–2A' and S2C).

In conclusion, *citk-1* and *citk-2* are suggested kinase-less orthologs of the neuronal isoform of mammalian CIT. Additionally, mutations in *citk-1* and *citk-2* can partially suppress the microtubule-dynamics-associated neuronal ectopic extension phenotypes of *klp-7(0)* mutant.

## CITK proteins regulate dynamic microtubules in the anterior process of PLM neurons and microtubule arrangement in the posterior process of PLM neurons

Building on our previous observations, we investigated the effects of loss of *citk-1* and *citk-2* on microtubule dynamics in the PLM mechanosensory neurons. We live-imaged microtubule plus-end dynamics using the established EBP-2::GFP reporter (plus-end binding protein) [2,68,29]. To quantify the dynamics of microtubules, we generated kymographs from time-lapse live images spanning 30 μm regions of interest in the anterior and posterior processes of PLM neurons (Fig 3A and 3C).

We observed that in comparison to wild-type animals, both *citk-1(0)* and *citk-2(0)* mutant animals display an increase in the number of growing microtubules/EBP-2::GFP comets in the PLM anterior process (Fig 3A and 3B). Moreover, the growth length and growth duration of these microtubule growth events significantly reduced (Figs 3A, S3A and S3B). This suggested an increase in the plus-end microtubule dynamics in the PLM anterior process. In contrast, the posterior process did not show a significant change in the number of growing comets (Figs 3C and S3D).

Furthermore, the arrangement of microtubules in the anterior and posterior processes of PLM neurons is well defined [2,29]. The anterior process typically exhibits a plus-end-out microtubule polarity, similar to vertebrate axons. The kymographs from this region display a majority of diagonal tracks directed away from the cell body (plus-end-out, green trace) (Figs 3A and S3C). In contrast, the posterior process is known to display a mixed microtubule polarity, similar to the vertebrate dendrites [2]. Hence, kymographs exhibit a comparable fraction of comets directed away from (plus-end-out, green trace) and towards the cell body (minus-end-out, purple trace) (Fig 3C and 3D). We observed that the mutations in neither *citk-1(0)* nor *citk-2(0)* perturb the plus-end-out microtubule polarity in the anterior process of PLM neurons (Figs 3A and S3C). However, both *citk-1* and *citk-2* mutant animals displayed an increase in the fraction of minus-end-out microtubules in the PLM posterior process, suggesting a shift towards predominantly minus-end-out polarity (Fig 3C and 3D).

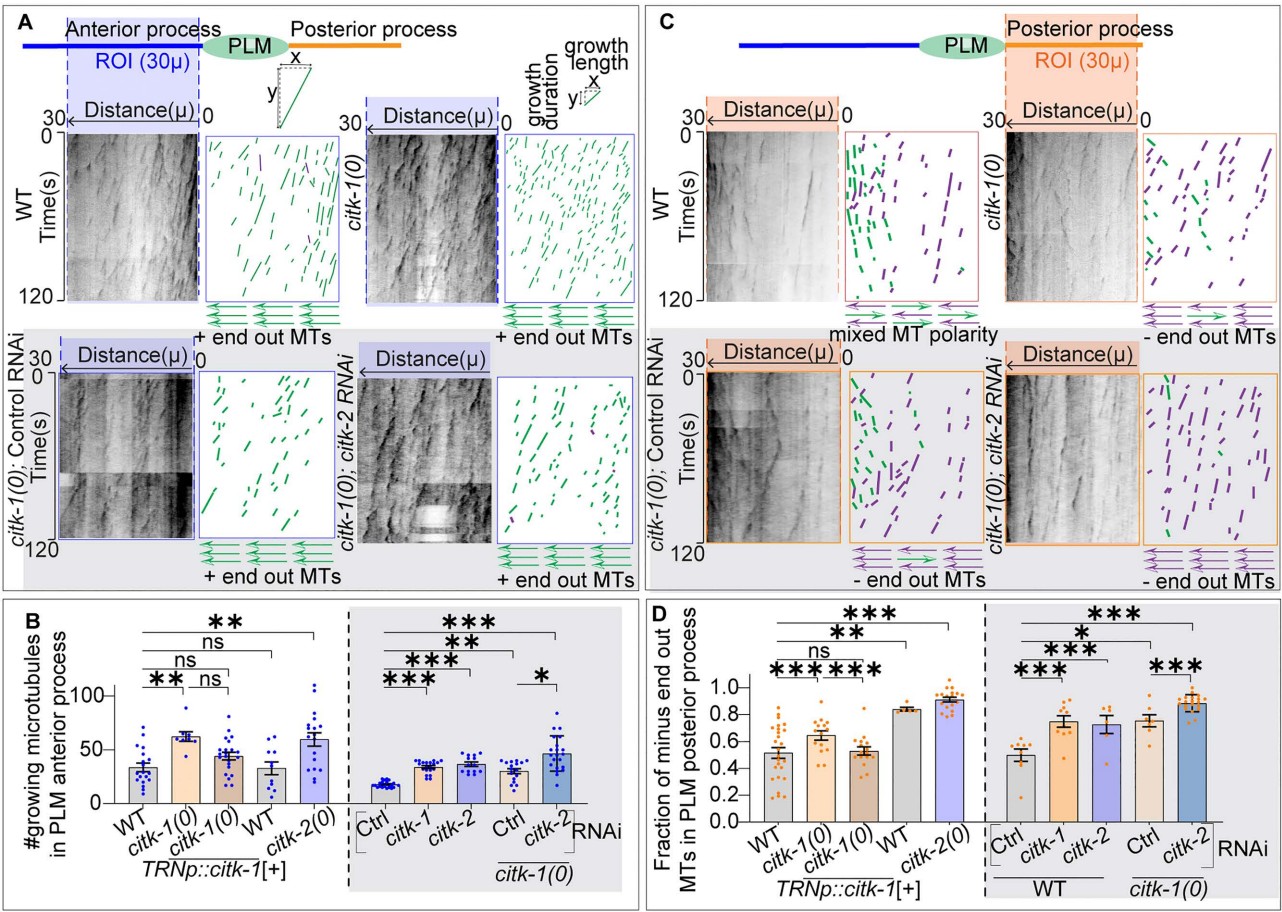

**Fig 3. CITK-1 and CITK-2 synergistically regulate microtubule dynamics in PLM neurons. (A, C)** Schematic representation of the PLM neuron and representative kymographs and schematics of the EBP-2::GFP dynamics obtained from the anterior (A) and posterior (C) processes of PLM neurons in the wild-type, *citk-1(0)* (white background) (acquired using ANDOR camera on a Nikon Ti2 fluorescence system), and RNAi strain (in gray background) *citk-1(0); lin-15b (-); shrsI2(pmec-3-sid-1 + unc-119[+])* fed on control and *citk-2 RNAi* bacteria (acquired on a Zeiss spinning disc confocal microscope). All these strains express *pmec-4*::EBP-2::GFP (*juIs338*) reporter. (The blue and orange dotted rectangles mark the 30um region of interest (ROI) in the (A) anterior (in blue) and (B) posterior process (in orange) of PLM neuron. These Rois are used to generate kymographs. One of the tracks from WT and *citk-1(0)* kymographs is enlarged in (A) to show its slope (ratio of growth length(x) and growth duration(y)) the direction of growth. The transgene *pmec-3-sid-1* is used to overexpress *sid-1* and enhance RNAi sensitivity in tissues expressing *pmec*-3, which includes PLM mechanosensory neurons. In the schematics, green and purple traces represent movements of EBP-2 bound microtubules away from the cell body (Plus end out) and movements towards the cell body (Minus end out), respectively. **(B, D)** Quantification of the number of observed diagonal tracks of EBP::GFP/growing microtubules in PLM anterior process (B) and fraction of minus-end-out microtubules (the number of 'minus-end out' microtubules/Total number of microtubules) in the posterior process of PLM neurons in different backgrounds. N = 3–4 independent replicates, n (number of neurons) = 14-63. The *shrEx512(pmec-4::citk-1(cDNA)[+])* is a transgene of *citk-1(cDNA)* expressed under *pmec-4*. N = 3 independent replicates, n (number of neurons) = 8-22. For C-D Error bars represent SEM (Standard error mean), ***, P < 0.001; **, P < 0.01, *, P < 0.05, ns, not significant. The groups separated by dotted lines were analyzed independent of each other. For (B) strains left of dotted line, P values from ANOVA with Tukey's multiple comparison test and for strains right of dotted line P values from Kruskal Wallis test followed by Dunn's multiple comparisons. For **(D)** P values from Brown Forsythe and Welch ANOVA followed by Dunn's multiple comparisons test.

The similar mutant phenotypes and molecular homology between *citk-1* and *citk-2* suggested a functional redundancy. To investigate the impact of losing both CITK-1 and CITK-2, we initially attempted to generate *citk-1(0); citk-2(0)* double homozygotes. However, the double homozygotes are not viable. This lethality is consistent with previous reports showing that simultaneous knockdown of both orthologues leads to gonadal sterility and embryonic lethality [66,69]. Consequently,

we employed a neuron-specific knockdown strategy by feeding *citk-2* RNAi to *citk-1(ok2328)* mutants in a strain with enhanced RNAi sensitivity in mechanosensory neurons [70–74]. We found that single knockdowns in the RNAi control strain mirrored the defects seen in single mutant animals (Figs 3A–3D and S3A–S3D). These defects were significantly aggravated by the knockdown of *citk-2* in the *citk-1* mutant background. In the PLM anterior process, the number of growing microtubules is significantly higher than in the single mutants (Fig 3A and 3C), and we observed an additive decrease in growth length and growth duration (Figs 3A, S3A and S3B). Furthermore, the posterior process of PLM neurons exhibited a significantly higher fraction of minus-end-out comets compared to single mutants, while the plus-end-out microtubule polarity of the anterior process remained unaffected (Figs 3A, 3C, 3D and S3C). These synergistic increases in microtubule dynamics confirm that CITK-1 and CITK-2 act redundantly to maintain neuronal microtubule polarity and stability.

We validated the cell-autonomous role of *citk-1* by expressing wild-type *citk-1*(cDNA) under the touch receptor neuron-specific promoter (*pmec-4)* in the *citk-1(0)* mutant animals. In these rescued animals, the number of growing microtubules in the PLM anterior process was comparable to wild-type animals, significantly less than the mutant animals (Fig 3B). Conversely, we observed an increase in both growth length and growth duration (S3A and S3B Fig). Moreover, the rescued neurons also restored mixed microtubule polarity, reversing the trend towards minus-end-out MT polarity in the mutant animals (Fig 3D). These findings demonstrate that mechanosensory neuron-specific expression of *citk-1* is sufficient to rescue the microtubule defects, confirming that CITK-1 functions cell-autonomously to regulate neuronal microtubule stability and polarity.

To further explore how *citk-1* regulates microtubule dynamics in PLM neurons, we overexpressed *citk-1* cDNA in mechanosensory neurons of wild-type animals. This resulted in an increased growth length and reduced growth duration of microtubules in the PLM anterior process without altering microtubule number or polarity (Figs 3B and S3A–S3C). This suggests that elevated levels of CITK-1 promote more rapid, sustained growth of existing polymerizing microtubules. Notably, the posterior process showed an increased fraction of minus-end-out microtubules upon overexpression (Fig 3D). This indicates that *citk-1* might regulate microtubule polarity in the PLM posterior process in a dosage-dependent manner, further supporting its role as a key regulator of MT polarity in the PLM posterior process.

We next investigated whether CITK-1 also influences the minus ends of microtubules in PLM neurons. Using the fluorescently tagged minus-end binding protein PTRN-1 (worm homolog of CAMSAP) [75,76], we quantified the number of *ptrn-1::*GFP puncta in PLM neurons of wild-type and *citk-1(0)* mutant animals (S3E Fig). We observed no significant change in anterior or posterior process of PLM neurons (S3E and S3F Fig). This suggests that CITK-1 neither regulates localization nor maintenance of this minus end marker in PLM mechanosensory neurons.

Mammalian CIT-N mediated actin modulation has been implicated in regulating dendritic morphogenesis and neurite extension [59,61]. Therefore, we analyzed the actin dynamics in anterior process of PLM neurons in *citk-1(0)* animals using a previously established GFP::UtCH reporter (Actin binding protein Utrophin) [77–79]. However, we did not observe any significant change in actin dynamics in PLM neurons of mutant animals (S3G and S3H Fig). This suggests worm *citk-1* is not involved in the modulation of actin dynamics in neurons.

Taken together, these observations indicate that *citk-1* and *citk-2* redundantly limit the microtubule plus-end dynamics in the PLM anterior process. Furthermore, they maintain the plus-end-out population of microtubules in the posterior process of PLM neurons, thereby ensuring the mixed microtubule polarity characteristic of this dendrite-like compartment.

**PLM mechanosensory neurons displayed various morphological defects on loss of both *citk-1* and *citk-2***

As previously described, perturbations in microtubule dynamics are associated with changes in the structure and function of these neurons, such as ectopic neurite growth observed in the *klp-7(0)* animals [2,14,29]. Therefore, to investigate whether the microtubule defects in *citk-1* and *citk-2* mutant animals translate to morphological changes, we analyzed PLM mechanosensory neurons in larval stage 4 animals. In wild-type animals, the PLM anterior process extends ventrolaterally

towards the midbody past the vulva landmark (marked by an asterisk, in Figs 4A and S4A–S4A'). A short posterior process extends into the tail (Figs 4A and S4A–S4A'). While we observed no significant overall alterations in PLM morphology in *citk-1* or *citk-2* single mutants, specific growth defects were evident in the posterior process (S4A and S4B Fig). Both the loss-of-function allele *citk-1(ok2328)* and the missense allele, *citk-1(shr11),* displayed a significantly shorter PLM posterior process compared to the wild-type animals, which can be rescued by extrachromosomal expression of the WT *citk-1* gene (S4A and S4B Fig). However, the length of the PLM anterior process in *citk-1* mutants was similar to the wild-type animals (S4C Fig). Furthermore, the ventral synaptic branch, which forms a little posterior to the vulva in wild-type animals (marked by a white arrow in Figs 4A and S4A), was misplaced or duplicated in 11% of *citk-1(ok2328),* 7% of *citk-1(shr11),* and 6% of *citk-2(ok2885)* mutant animals (S4D Fig).

Given the redundant roles of these proteins in microtubule regulation, we investigated the impact of their simultaneous loss on PLM morphology. Using neuron-specific knockdown of *citk-2* in *citk-1(ok2328)* mutant animals, we observed weakly-penetrant growth defects in the PLM mechanosensory neurons of these animals.

As illustrated in two panels of Fig 4A (Inset-2), the loss of *citk-2* further declines the length of the posterior process in *citk-1(0)* animals (Fig 4A and 4B). This synergistic reduction suggests that precise extension of this dendritic-like compartment is regulated by both *citk* orthologs. In contrast, in *citk-1(0) citk-2(kd)* animals most of the PLM anterior processes (80 out of 97) displayed wild-type like growth, terminating near or before the ALM cell body (Fig 4A, inset-1). Despite the overall stability of the anterior process, we observed a small fraction of neurons with neurite growth defects, including overgrowth past the ALM cell body or premature termination before vulval opening (short neurite defect) (Fig 4A and 4C). However, quantitative analysis revealed no significant difference in the average length of PLM anterior processes between mutant and wild-type animals (S4E Fig).

Furthermore, we observed irregularities in synaptic branch formation in *citk-1(0); citk-2(kd)* animals. The synaptic branch was either missing or mislocalized in a fraction of PLM neurons. (Fig 4A and 4D). Additionally, these animals displayed ectopic protrusions emanating from the PLM cell body and anterior as well as posterior processes (S4F Fig).

The observed structural defects, such as ectopic protrusions and growth defects, are markers of altered microtubule dynamics in the PLM neurons [35,38,45]. Collectively, these findings reinforce the hypothesis that CITK-1 and CITK-2 jointly regulate the microtubule dynamics and, hence, the development of PLM neurons. However, the precise mechanism of this regulation remains unclear.

## Microcephaly-associated protein ASPM-1 phenocopies the loss of citron kinase proteins CITK-1 and CITK-2

Previous molecular studies have shown that mammalian citron kinase is recruited to spindle poles during cytokinesis by ASPM, a microtubule minus-end-associated protein. It regulates the organization and dynamics of spindle microtubule arrays [56,80]. Additionally, both proteins have been shown to colocalize to the midbody region during cytokinesis [81]. Similarly, the worm orthologue of ASPM, ASPM-1, localizes to the spindle poles and is crucial for spindle pole organization. Consistent with these roles, protein network analysis [82] identified strong predicted interactions between *aspm-1* and *citk-1* as well as *citk-2* (Fig 5A). Drawing on these findings, we hypothesized that citron kinase interacts with ASPM-1 in neurons to regulate microtubule dynamics.

Previous RNAi studies have associated *aspm-1* knockdown with embryonic sterility and lethality, similar to the loss of both *citk-1* and *citk-2* [83]. Therefore, we utilized a temperature-sensitive variant of *aspm-1* [84] to characterize its role in regulating microtubule dynamics in neurons. Using the EBP-2::GFP reporter, we observed that *aspm-1(ts)* mutants exhibit microtubule defects remarkably similar to those seen in *citk* mutants. Specifically, the PLM anterior process showed a significant increase in the number of growing microtubules (EBP-2::GFP comets) (Fig 5B and 5D). The growth length and duration of these microtubule growth events were significantly reduced compared to wild-type animals (Figs 5B, S5A and S5B). In contrast, PLM posterior processes displayed no change in the number of growth events consistent with the *citk* mutant phenotype (Fig 5C). Additionally, although the anterior processes showed no change in microtubule polarity (Figs

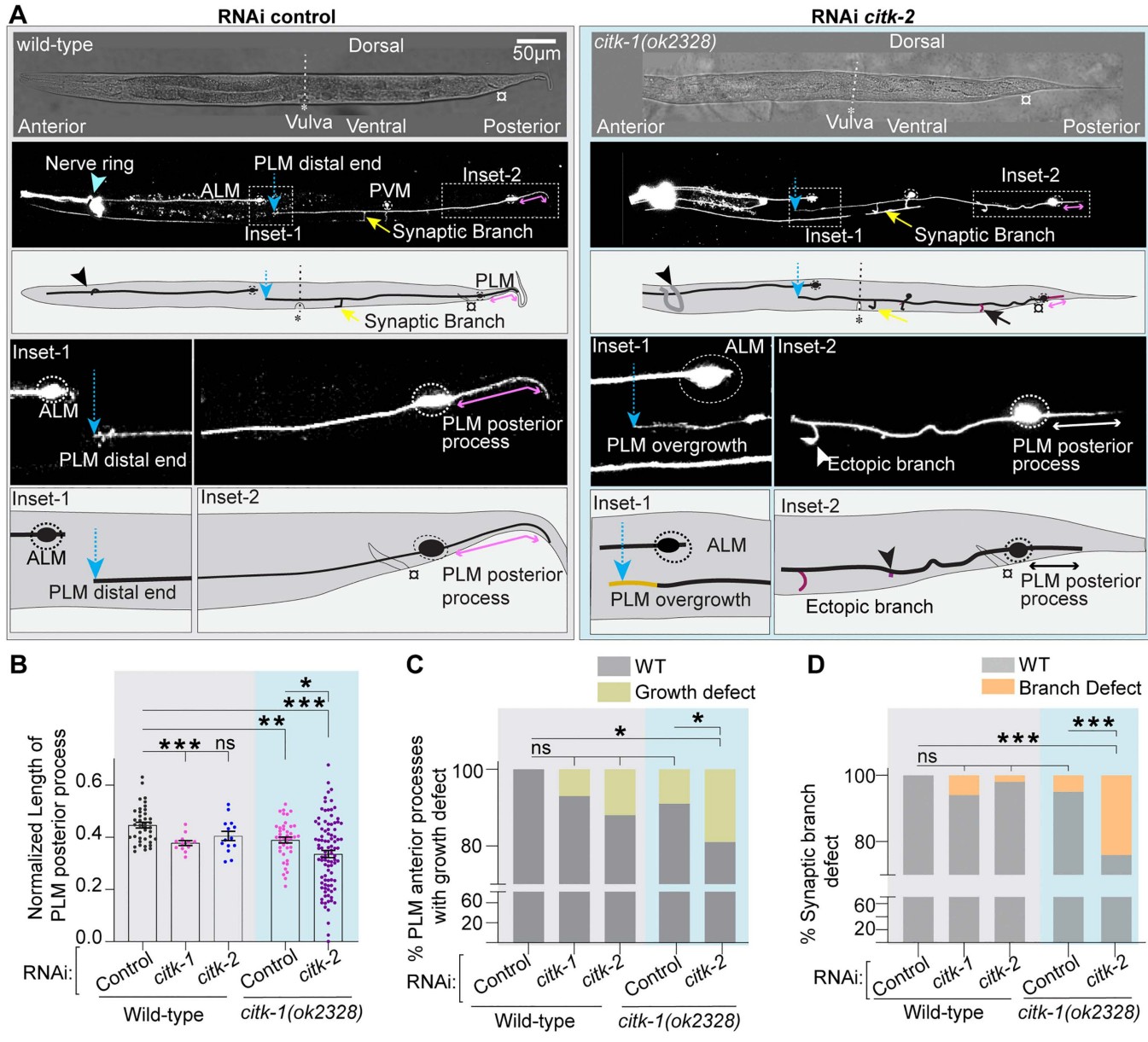

**Fig 4. Growth and guidance defects observed on knockdown of *citk-2* in *citk-1(0)* animals. (A)** Representative confocal images and schematics of ALM and PLM mechanosensory neurons of L4-stage WT animals fed on empty vector(control) and *citk-1(ok2328)* mutant animals fed on control *citk-2* dsRNA. Both animals have an RNAi sensitive background *lin-15b(-); shrsl2(pmec-3-sid-1 + unc-119[+])*. The head of the worm (anterior end) is placed towards left, the tail (posterior end) points to the right, the dorsal edge of its body faces upward and the ventral edge with vulva faces downwards in the images. The confocal image is split into DIC channel image (top panel) and fluorescent image (lower panel) image. The mechanosensory neurons in these animals are labeled by a *pmec-4::mCherry (tbIs222)* reporter. Inset-1 zooms in on the distal end (marked by a blue dotted arrow) of anterior process of PLM neuron and the location of ALM cell body. Inset-2 focuses on the extent of PLM posterior process (marked by a double-headed pink arrow). The yellow arrow points to the ventral synaptic branch and the white arrowhead marks the ectopic branch. Vulva is marked by an asterisk (*), anus is marked by a generic symbol (¤) and the celeste-blue arrowhead points to the nerve ring. **(B)** Quantification of the normalized length of PLM posterior process in the wild-type control and *citk-1(0)* RNAi sensitive background animals fed on bacteria expressing either empty vector (control) or *citk-1* or *citk-2 ds* RNA for RNAi mediated knockdown. **(C)** Plot of percentage of PLM neurons exhibiting anterior neurite growth defects such as overgrowth of PLM neuron (marked in light brown in inset-1 **(A)**). **(D)** Quantification of percentage of PLM neurons displaying a ventral synaptic branch defect, which in wild-type animals is localized a little posterior to vulva (labeled by yellow arrow in **(A)**). In mutant animals it can be mispositioned or duplicated or completely absent. B-D, N = 3 independent replicates, n (number of neurons) = 14-96. For B-D, ***, P < 0.001; **, P < 0.01, *, P < 0.01; ns, not significant. For B, P values from Brown Forsythe and Welch ANOVA test. Error bars represent SEM (Standard error mean). For C-D, P values from Fisher's exact test for WT phenotype and defect phenotype.

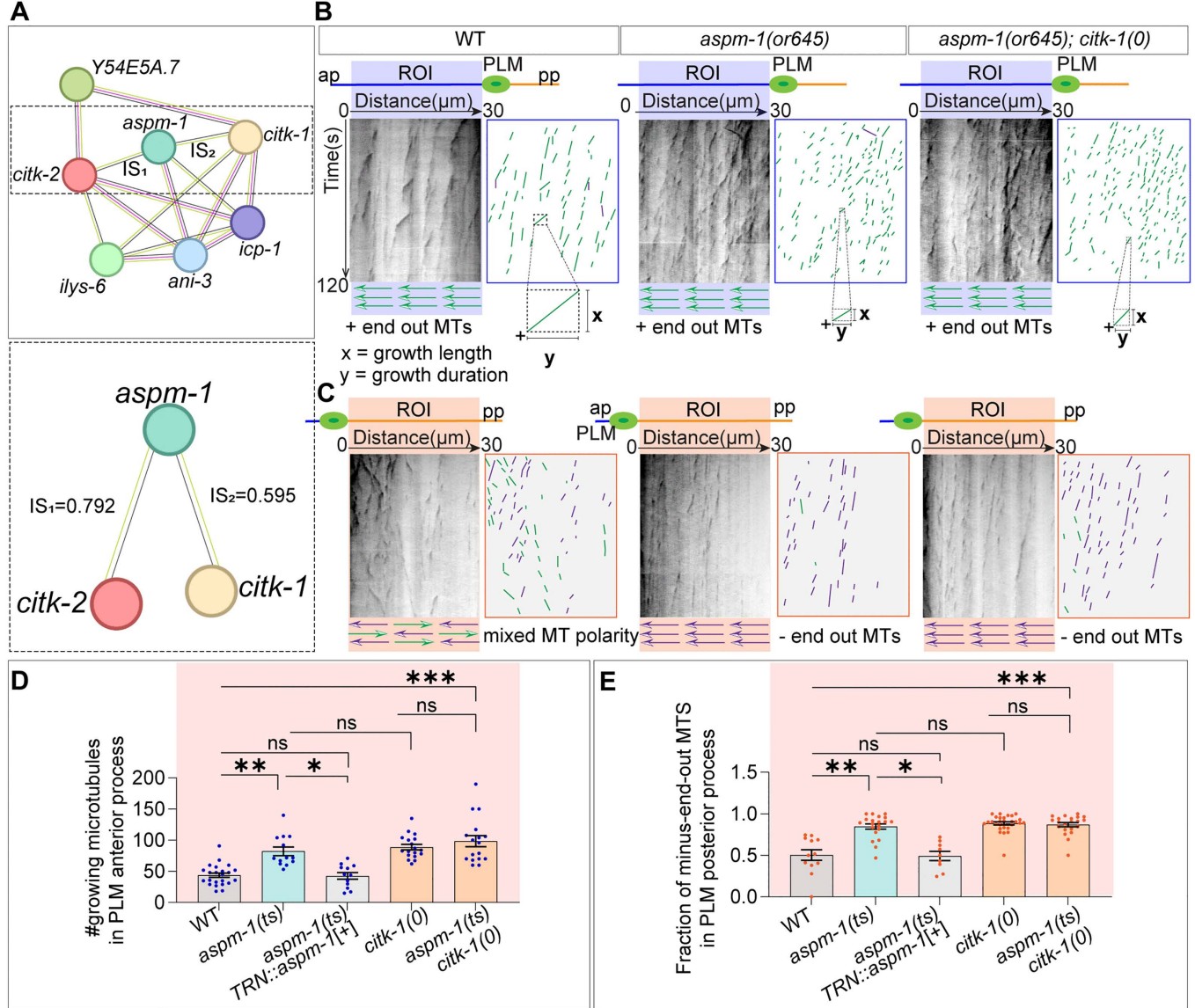

**Fig 5. Microtubule dynamics and fractional polarity defects observed in *aspm-1(ts)* mutants. (A)** Protein-protein interaction network obtained for *citk-1* and *citk-2* using STRING database. The nodes represent query and known interacting proteins. Edges represent known and predicted protein-protein functional associations (not necessarily physical interaction) between these proteins or their putative homologs in varied species. The type of association is color coded based on evidence for association such as: Magenta edges (○—○): experimentally determined association based on bio-chemical or genetic data (including experimental evidence for interaction of putative homologs in other species), Black edges (○—○): co-expression, green edges (○—○): text mining (Co-mentioned in PubMed abstracts). 'IS' is the combined interaction score which represents an approx-imation of confidence in a given association being true, based on all available evidence, on a scale of zero to one. An interaction score of 0.5 or less indicates 50% incidence of false positives. CITK-1 and CITK-2 have no known direct physical association, however, both show interaction with ASPM-1 with a confidence score >0.5 **(B-C)** Representative kymographs and schematics obtained by plotting the movement of EBP-2::GFP (*juIs338*) comets in the anterior (B) and posterior (C) processes of PLM neurons (from ROIs mentioned in Fig 3) of the wild-type, *aspm-1(ts)* and *aspm-1(ts); citk-1(ok2328)* mutant animals grown at non-permissive temperature of 25°C. (ts) represents temperature sensitive allele of *aspm-1*. The diagonal lines represent the number of EBP-2::GFP tagged plus-end of microtubules growing away from (green) or towards (purple) the PLM cell body. The slope of the line is the ratio of the growth length(x) and growth duration(y) of a polymerization event. One of the tracks is magnified to compare the growth lengths and growth durations between different genotypes in the anterior process. **(D-E)** Quantification of the number of growing microtubules/EBP-2::GFP comets in the PLM anterior process (D) and the fraction of minus-end out microtubules in the PLM posterior process (E) in different backgrounds grown at non-permissive temperature. N = 3-4 biological replicates, n (number of neurons) = 12-23. Error bars represent SEM (Standard error mean). For D, E, ***, $P < 0.001$; **, $P < 0.01$; *, $P < 0.05$; ns, not significant. P values from Kruskal-Wallis test followed by Dunn's multiple comparison test.

5B and S5C), the PLM posterior processes displayed a significant increase in the fraction of minus-end-out growing micro-tubules (Fig 5C and 5E), suggesting a shift in microtubule polarity in *aspm-1(ts)* mutants similar to that of *citk* mutants. These defects were rescued by transgenic expression of *aspm-1*(cDNA) under the touch neuron-specific promoter *pmec-4,* suggesting a cell-autonomous role for *aspm-1* in neuronal microtubule regulation (Figs 5D, 5E, S5A and S5B).

Moreover, when we overexpressed the *pmec-4::aspm 1*(cDNA) transgene in wild-type animals, we observed an increase in microtubule growth length and growth duration in the PLM anterior processes (S5H and S5I Fig). Notably, the fraction of minus-end-out microtubules in the posterior process increases significantly (S5J Fig), mirroring the effects of *citk-1* overexpression in wild-type animals (Fig 3D). This suggested role of ASPM-1 in a sustained plus-end growth in the PLM anterior process and dose-dependent regulation of microtubule polarity in PLM posterior process.

Therefore, we examined the microtubule plus-end-dynamics in *aspm-1(ts); citk-1(0)* and *aspm-1(ts); citk-2(0)* double mutants to evaluate genetic interaction (Fig 5B and 5C). We observed no additive changes in microtubule dynamics or polarity in PLM neurons of these animals, suggesting that *aspm-1* works in same genetic pathway as *citk-1* (Figs 5D, 5E and S5A–S5C) or *citk-2* (S5D–S5G Fig).

To further validate that ASPM-1 plays a post-mitotic developmental role in the microtubule regulation of PLM mechanosensory neurons, we subjected *aspm-1(ts)* mutant animals to the non-permissive temperature from the L1 growth stage (growth condition-II, S6A Fig), after the PLM neurons have already differentiated [43]. The defects in microtubule dynamics displayed by these animals were comparable to the F1 progeny *aspm-1(ts)* mutant animals who remained at the non-permissive temperature through embryonic development (growth condition-I, S6A Fig) (S6B–S6D Fig). These animals displayed similar shift towards minus-end-out microtubule polarity in PLM posterior process as well (S6G Fig). This suggests that *aspm-1* is required post-mitotically for the maintenance of MT dynamics and polarity in differentiated PLM neurons (S5D–S5G Fig).

Building on the observations that *aspm-1* regulates microtubule dynamics in the PLM mechanosensory neurons, similar to citron kinase mutants, we further evaluated whether it could suppress the *klp-7(0)* ectopic extension phenotypes [2,29]. As expected, the introduction of *aspm-1(ts)* mutation in the *klp-7(0)* mutant background partially suppresses the ALM ectopic extension phenotype (S7A and S7B Fig).

Additionally, previous *in-vitro* studies have shown that ASPM colocalizes with the minus-end binding protein CAMSAP (PTRN-1 homolog) [85,86]. Hence, we investigated whether microtubule minus-end-dynamics or PTRN-1 puncta were perturbed in the PLM neurons of *aspm-1(ts)* mutant. We found that compared to wild-type animals, *aspm-1(ts)* mutants showed no change in the number of PTRN-1::GFP puncta in either the anterior or posterior processes of PLM neurons (S7C and S7D Fig). This lack of disruption suggests that ASPM-1 is not required for the proper localization or mainte-nance of these microtubule minus-end markers.

These observations suggest that ASPM-1 works in the same genetic pathway as CITK-1 and CITK-2 to differentially regulate microtubules in the anterior and posterior processes of post-mitotic PLM neurons. Specifically, ASPM-1 regulates microtubule stability in the anterior process and maintains the population of plus-end-out microtubules in the posterior process post-mitotically.

## CITK-1 and ASPM-1 colocalize with microtubule plus-end binding protein

Because both CITK-1 and ASPM-1 influence the microtubule plus-end dynamics but not the minus-end dynamics, we investigated whether these proteins localize to the growing plus ends. Using a TRN-specific promoter, we transgenically co-expressed fluorescent reporters of either CITK-1 or ASPM-1 with the plus-end-binding protein EBP-2 (Fig 6A and 6C). We observed punctate localization of both CITK-1 and ASPM-1 in the cell body and the anterior and posterior processes of PLM neurons (Fig 6A and 6C). Multichannel fluorescence intensity plots revealed that GFP::CITK-1 puncta frequently trail the EBP-2::*mCherry* peaks (Fig 6A and 6B and S1 Video). Pearson's correlation analysis confirmed a moderate posi-tive correlation between GFP::CITK-1 and EBP-2::*mCherry* (Fig 6E), with a similar correlation observed for GFP::ASPM-1

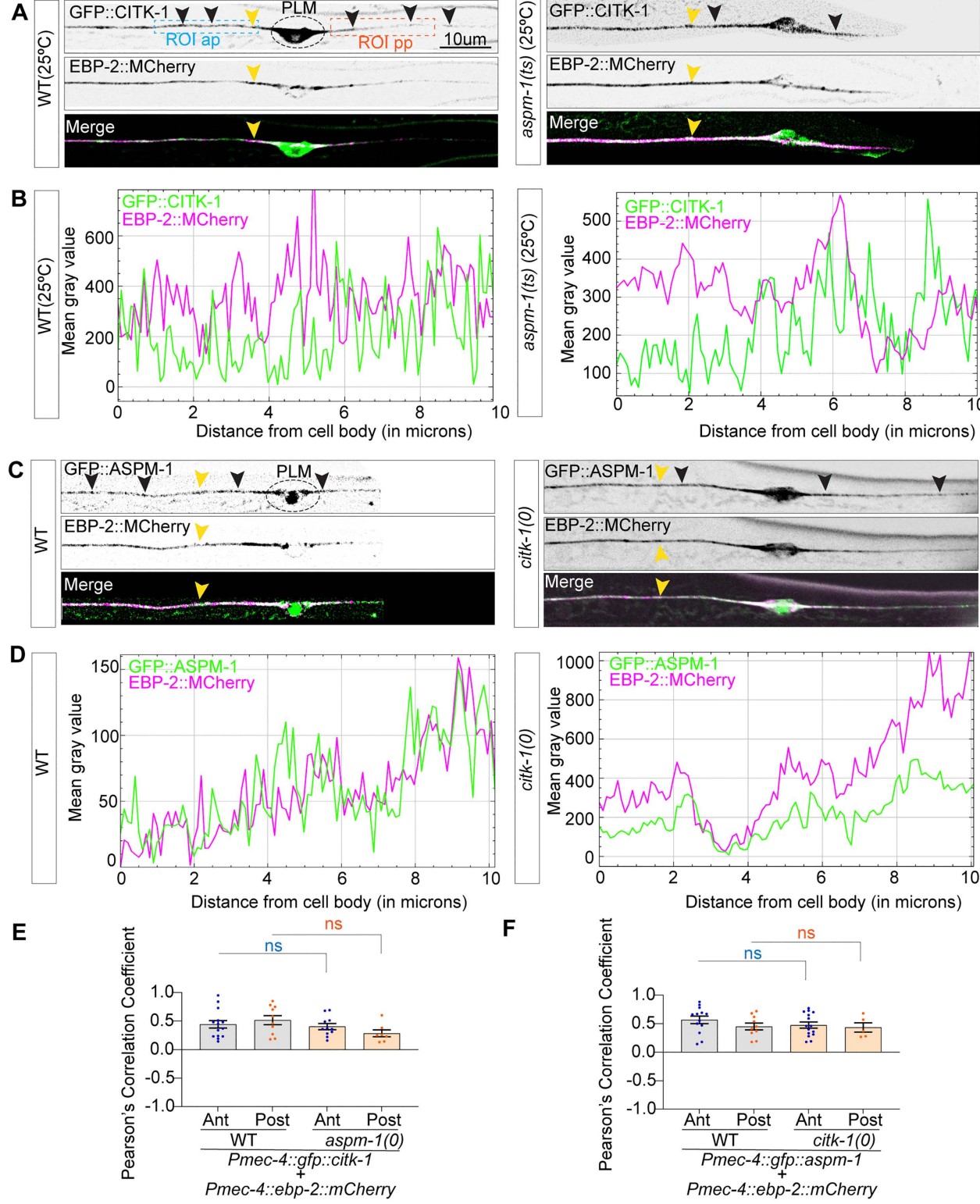

**Fig 6. Localization of CITK-1 and ASPM-1 in PLM neurons displays moderate correlation with MT plus-end protein, EBP-2. (A)** Representative confocal images of the PLM mechanosensory neurons in Larval stage-4 animals expressing *shrEx550 (pmec-4::gfp::citk-1* + p*mec-4::ebp-2::mCherry)*

in wild-type and *aspm-1(ts)* mutant background. **(B)** Representative multichannel plot profile depicting mean intensity value of GFP::CITK-1(depicted in green) and EBP-2::MCherry(depicted in magenta) along the 10um ROI in the anterior process of PLM neuron in wild-type and *aspm-1(ts)* background. **(C)** Confocal images depicting PLM mechanosensory neurons in Larval stage-4 animals expressing transgene *shrEx552 (pmec-4::gfp::aspm-1+pmec-4::ebp-2::mCherry)* translational reporters in wild-type and *citk-1(0)* backgrounds. **(D)** Representative multichannel plot profile depicting mean intensity value of GFP::ASPM-1(depicted in green) and EBP-2::MCherry(depicted in magenta) along the 10um ROI in the anterior process of PLM neuron in wild-type and *citk-1(0)* background. **(E-F)** Quantification of Pearson's correlation coefficient between GFP::CITK-1 and EBP-2::MCherry in WT and *aspm-1(ts)* backgrounds grown at non-permissive temperature and **(E)** GFP::ASPM-1 and EBP-2::MCherry in WT and *citk-1(0)* animals (F) from 30um regions of interest in PLM anterior and posterior process similar to ROIs used in Fig 3. For E, N = 2-3 biological replicates, n (number of neurons) = 6-14. For F, N = 2-3 biological replicates, n(number of neurons) = 5-15. For E and F, Error bars represent SEM (Standard error mean), \*\*\*, $P < 0.001$; \*\*, $P < 0.01$; \*, $P < 0.05$, ns, not significant. P values from ANOVA with Tukey's post hoc comparison. Ns, not significant. For Pearson coefficient Correlation coefficient (r) varies from 0.8-1.

and EBP-2::*mCherry* (Fig 6C, 6D, 6F and S2 Video). These data suggest a plus-end association for both CITK-1 and ASPM-1 within PLM processes.

Given that *citk-1* and *aspm-1* function in the same genetic pathway, we tested whether their plus-end localization is mutually dependent. We compared the CITK-1 and EBP-2::*mCherry* correlation between wild-type and *aspm-1(0)* animals (Fig 6A). We observed no significant changes in the localization pattern or Pearson's correlation coefficient (Fig 6B and 6E). Similarly, the correlation between ASPM-1 and EBP-2::*mCherry* remained unchanged in *citk-1(0)* mutant animals (Fig 6C, 6D and 6F).

These observations suggest that CITK-1 and ASPM-1 are associated with and regulate microtubule plus-end dynamics in neurons. However, unlike dividing cells, the localization of CITK and ASPM-1 is independent of each other in the PLM mechanosensory neurons.

## Discussion

In this study, we identified *W02B8.2/citk-1,* suggested kinase-less orthologue of mammalian citron-rho interacting kinase (*CIT*), through a genetic modifier screen designed to isolate novel regulators of neuronal microtubule dynamics. Along with its paralog *F59A6.5/citk-2, citk-1* plays a redundant and essential role in the development of PLM neurons. These proteins regulate plus-end dynamics in the axon-like anterior process and maintain microtubule polarity in the dendrite-like posterior process of PLM neurons (Fig 7A and 7B). Our findings further demonstrate that the *citk-1/2* operate in the same genetic pathway as *aspm-1* to modulate microtubule dynamics. Notably, both CITK-1 and ASPM-1 exhibit a positive correlation with the microtubule plus-end binding protein EBP-2 in PLM mechanosensory neurons of *C. elegans*. Intriguingly, unlike the recruitment hierarchy observed during cell division, the localization of CITK-1 and ASPM-1 at these plus-ends is mutually independent (Fig 7). This suggests that these cytokinesis-associated factors have been repurposed into a specialized regulatory module to maintain the structural integrity of differentiated neurons.

### CITK-1 and ASPM-1 maintain axonal microtubule stability in PLM neurons

Originally identified as a Rho/Rock effector, Citron-rho interacting kinase(CITK) is a well-established regulator of cytokinesis, specifically through the organization of spindle microtubule arrays and midbody maturation [51]. This function is evolutionarily conserved; in *C. elegans,* the two suggested CIT orthologues, *W02B8.2/citk-1* and *F59A6.5/citk-2,* function redundantly to ensure developmental viability [66]. Indeed, our attempts to generate viable double homozygotes were unsuccessful, aligning with the reports that their combined loss leads to gonadal sterility and embryonic lethality.

Structurally, we found that both CITK-1 and CITK-2 are homologs of the mammalian CIT-N isoform, which specifically lacks the N-terminal kinase domain [51,67]. CIT-N has been shown *in vitro* to localize in dendrites and soma of hippocampal glutamatergic neurons and GABAergic and glutamatergic neuronal populations of other brain areas, including thalamus, neocortex, and specific populations of cerebellar Pyramidal neurons [51,60]. It has been shown to regulate spine

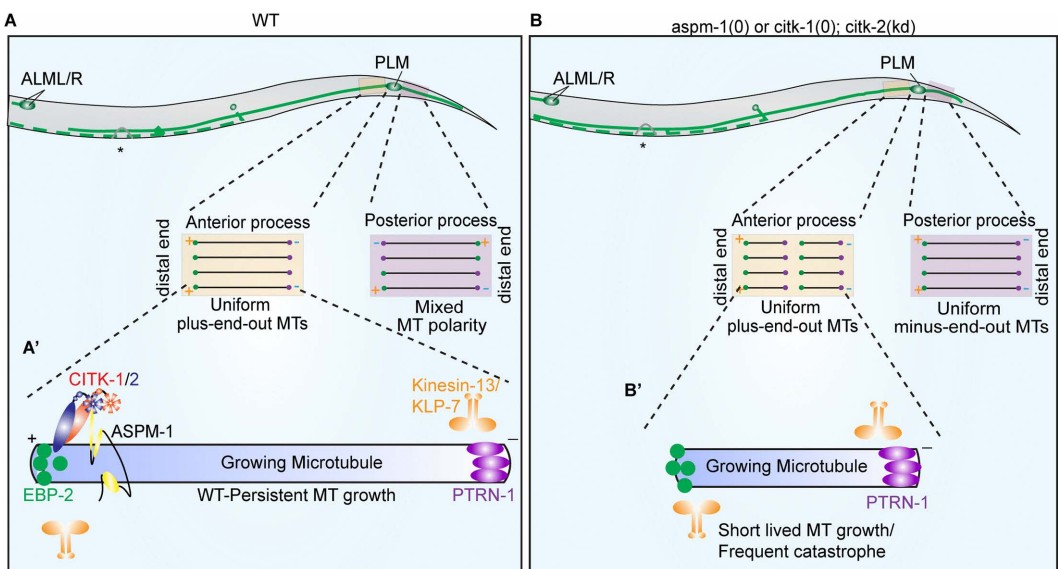

**Fig 7. Summary and Proposed model.** Schematic representation of microtubule dynamics in the anterior and posterior processes of PLM mechano-sensory neurons, in the wild-type *C. elegans* animals (A) and the mutant animals **(B)**. The 30 μm ROI in the anterior process (orange rectangular selection) and posterior process (pink rectangular selection) are zoomed-in to illustrate microtubule dynamics in these processes. The black lines represent microtubules, the sold green circle depicts the growing plus-ends of microtubules, bound by plus-end-binding proteins and the solid purple circle depicts the minus-ends of microtubules. The direction of arrowhead shows the direction of microtubule growth. **(A)** In the wild-type animals the anterior process exhibits a plus-end-out steady-state microtubule dynamics, while the posterior process displays a mixed microtubule polarity. **(B)** Similar ROI in mutant animals shows increased microtubule dynamics in the anterior process, schematized as increased number of growing microtubules, and a minus-end-out microtubule polarity in the posterior process. (A'-B') Schematic representation of microtubule plus end localization of CITK-1 and ASPM-1 protein trailing the plus end binding protein EBP-2. CITK-1 and ASPM-1 promote MT plus-end growth (A'). In the absence of CITK-1 and/or ASPM-1 microtubule plus-ends are more dynamic and are probably exposed to destabilization by KLP-7(Kinesin-13 family protein) (B').

maturation and dendrite morphology *in-vitro* and *in vivo* [60,87]. Simultaneous knockdown of both CITK and CITN has also been shown to restrict axon outgrowth in cultured primary hippocampal neurons. CITK knockdown also blocks regeneration in DRGN neurons *in-vitro* [58]. The regulation of dendritic morphogenesis and neurite extension by the CIT protein has been attributed to Rho mediated modulation of actin [59,61]. Our *in vivo* findings in PLM neurons expand this understanding. We find that loss of both *citk-1* and *citk-2* results in shortening of the dendrite-like posterior process and growth defects in the axon-like anterior process of PLM neurons. However, we did not find any major change in actin dynamics in the anterior process of PLM neurons in *citk-1(0)* mutants. Furthermore, unlike the primarily dendritic localization reported in hippocampal neurons, we found that CITK-1 localizes to the soma and both neuronal processes, suggesting a broader, more integrated role for kinase-less CITK in the structural maintenance of differentiated PLM neurons.

While previous studies in dividing HEK cells established that citron kinase regulates the nucleation, stability, and length of astral microtubules [56], our work reveals a specialized post-mitotic role for the kinase-less orthologue in neurons. We found that CITK-1 specifically modulates the microtubule plus-end dynamics in the PLM anterior processes without altering the microtubule polarity in PLM anterior process or the recruitment or maintenance of minus-end out-associated protein PTRN-1. The moderate colocalization we observed between CITK-1 and the plus-end binding protein EBP-2 suggests that CITK-1 selectively regulates the population of dynamic microtubules in PLM neurons. Although the exact molecular mechanism by which CITK confers microtubule stability in the anterior process remains unclear, its presence at the plus-ends suggests it may function as a specialized scaffold or protective cap against depolymerization forces. This model is strongly supported by our genetic evidence, where the loss of *citk-1* partially suppresses the ectopic extension phenotype typically observed in *klp-7(0)* mutants. This suppression implies a functional antagonism between the stabilizing role of

CITK and the depolymerizing activity of KLP-7, highlighting a critical balance required for maintaining proper neuronal architecture.

During cytokinesis, CITK is specifically enriched in the midbody region through its interactions with the Rho GTPase pathway [ 53,88–91]. Additionally, along with its interacting partners, such as Centralspindilin complex and KIF14, CITK regulates cross-linking and bundling of microtubules to stabilize the midbody architecture [54,92–94]. Our findings raise the intriguing possibility that in the absence of a kinase domain, the worm CITK orthologs retain a functionally analogous scaffolding role in post-mitotic neurons, as the primary drivers of microtubule stabilization.

The clinical relevance of these findings is underscored by the association between mutations in mammalian citron kinase and MCPH (autosomal recessive primary microcephaly), a neurodevelopmental disorder characterized by reduced brain size and mild to severe delayed cognitive development and behavioral challenges [48,50]. From studies in CITK-/- mice, these defects were attributed to defects in division of neuronal precursor cells, apoptosis and cell migration [51,95]. In this study we find that loss of citron kinase leads to perturbation in microtubule dynamics and neurite growth in differentiated PLM neurons. These findings suggest that microtubule instability within post-mitotic neurons may be a significant yet understudied contributing factor to the neurological pathologies associated with MCPH.

Interestingly, more than 40% of MCPH-associated variants occur within the locus of ASPM, a microtubule minus-end-associated protein. The role of ASPM-1 in meiotic spindle rotation in *C. elegans* oocytes [84,96], and in symmetric divisions, cell migration and cell fate acquisition in neuronal precursor cells are documented [80,56,81,97,98,99,100,101]. However, its role in differentiated neurons has not been characterized.

In dividing HEK cells, ASPM is known to recruit citron kinase to spindle poles, where CITK acts as a vital downstream effector in regulating spindle array dynamics [56]. Our findings demonstrate that ASPM-1 and CITK-1/2 function in the same genetic pathway to limit the number of dynamic microtubules in the axon-like anterior process of PLM mechanosensory neurons. We also find a functional antagonism between ASPM-1 and KLP-7 in the regulation of neurite extension in ALM neurons. By identifying these effects in differentiated cells, we provide evidence that ASPM-1, long considered a mitotic factor, is repurposed in post-mitotic neurons to maintain neuronal microtubule dynamics. This is consistent with an emerging body of literature identifying cytokinesis-associated proteins with essential roles in post-mitotic neurons [102].

## CITK and ASPM-1 maintain the plus-end-out microtubule population in the dendrite like posterior neurite of PLM neuron

While ASPM, similar to CAMSAP, has been shown to regulate microtubule minus-end dynamics *in-vitro* [85], our *in vivo* study reveals a distinct role for the worm orthologue, ASPM-1. We found that ASPM-1, much like CITK-1 and CITK-2, regulates the population of plus-end-out microtubules in the dendrite-like posterior process of PLM neurons. This is in sharp contrast to PTRN-1/CAMSAP, which specifically regulates the minus-end-out population in the same compartment [2].

We found that ASPM-1 and CITK-1/2 operate within the same genetic pathway to govern microtubule polarity in the posterior process of PLM neurons. This is consistent with previous findings in dividing cells, where CIT-K works downstream of ASPM in regulating spindle arrays [56]. Previous studies have also shown that the C-terminal of ASPM coimmunoprecipitated with different myc-tagged CITK fusion proteins, including kinase-active (FL) or kinase-dead (KD) full-length CITK, citron-N (CN), or the N-terminal or C-terminal half of Citron kinase protein [56]. Furthermore, it has been suggested that the N-terminus of the ASPM protein localizes to centrosomes and spindle poles, and the C-terminus tagged with GFP localizes at the midbody region, enriched with antiparallel arrays of microtubules [81]. This suggests that ASPM can interact with CIT protein irrespective of the kinase domain. Interestingly, we found that both proteins show moderate colocalization with the plus-end binding protein EBP-2. However, their plus-end localization is mutually independent. These findings suggest that while ASPM and CITK epistatically regulate the microtubules in PLM neurons, the mechanistic basis for their recruitment in post-mitotic neurons differs from the hierarchical recruitment observed during cytokinesis.

In summary, this study identifies two microcephaly-associated proteins, kinase-less Citron Kinase (CITK-1/2) and ASPM-1, which are functionally repurposed in post-mitotic PLM mechanosensory neurons. While traditionally recognized for their role in cytokinesis, we demonstrate that these factors play a crucial role in regulating microtubule dynamics and polarity, thereby ensuring proper neuronal development.

## Materials and methods

### *C. elegans* genetics

*C.* elegans strains were reared on the standard NGM (nematode growth medium plates) seeded with the *E. coli* OP50 bacterial strain. For all experimental purposes, worms were maintained at 20°C, with the exception of the temperature-sensitive(ts) *aspm-1* strain [103]. For this strain P0 parents were shifted to non-permissive temperature (25°C) at the L4 stage, and their F1 progeny were imaged at the L4 stage. To investigate the post-mitotic role of ASPM-1 in microtubule regulation, L1 larvae were shifted to 25°C and imaged at the L4 stage, thereby bypassing the embryonic loss of ASPM-1 in differentiating PLM neurons. These two paradigms were distinguished as Growth Condition-I and Growth Condition-II, as illustrated in S6A Fig. All experimental controls were also subjected to identical temperature shifts.

The (0) denotes the loss of function allele; for instance, the *tm2143* deletion allele of *klp-7* is mentioned at places as *klp-7(0)*. The strains studied in this paper, including the mutants generated by EMS mutagenesis, are listed in S4 Table. The published transgenes used in this study are listed in S4 Table. The newly generated transgenic strains carrying extrachromosomal arrays of DNA are tabulated in S5 Table. The transgenic lines were generated by microinjection into the gonad of the parent [104]. The homozygosity of mutants was confirmed either by PCR genotyping or by sequencing.

### EMS mutagenesis

The suppressor mutants were generated by EMS mutagenesis [103]. The age-synchronized larval-4 stage animals were pooled and washed thrice with the 1xM9 solution (isotonic salt solution), to get rid of any gut bacteria, following which, the animals were incubated in the 1xM9 solution containing 47mM EMS at 20°C for 4 hours on a hula mixer, which allowed continuous mixing and aeration of media. Post-incubation, the animals were pelleted, and the supernatant was discarded, followed by three washes with 1XM9 solution. Post-wash animals were pelleted, and 6–7 healthy animals were transferred to fresh NGM plates seeded with OP50 bacteria. The resulting F1s were transferred to fresh plates and the F2 generation animals were screened for suppression phenotype. Only one suppressor was picked from one F1 plate. Interestingly, in all cases, the suppression phenotype in one F1 plate was shown by about 25% of the F2 animals, suggesting that the EMS-induced mutations were recessive.

The screening was conducted in three batches, as tabulated in S1 Table.

### Whole genome sequencing mapping analysis

The suppressor mutants were backcrossed with the wild-type N2 Bristol strain animals four times [105,106]. In the first three backcrosses, one F3 animal homozygous for *klp-7(tm2143)* mutation showing the suppression of ectopic extension phenotype similar to the original suppressor mutant was selected. In the fourth backcross, five to ten F3 animals were selected, one from each F2 plate to ensure the selection of unique recombinant animals in this step.

The progenies of the homozygous recombinant animals were pooled together to extract the genomic DNA by the isopropanol precipitation method [107]. The library preparation and Illumina sequencing of genomic DNA was outsourced to the Genome Technology Access Centre at McDonnell Genome Institute (MGI), Washington University. They constructed KAPA PCR-free libraries for each sample and sequenced on 0.02 total of a NovaSeq S4 flowcell (300 XP) to obtain paired-end Fastq files.

We analyzed the whole genome sequencing data using the tools available on the online Galaxy platform (listed in S6 Table) [108–111]. The default settings were used for the analysis with each tool. The mapping by sequencing guide at https://mimodd.readthedocs.io/en/latest/nacreousmap.html [109,111] was used to design the workflow for the analysis. The tools in S6 Table are listed in the order they were used for analysis.

## Sequencing of *citk-1(ok2328)* allele

For sequencing the mutant allele, we reverse transcribed the RNA isolated from the mutant animals. As previously described the about to be starved plates of mutant animals, which contained animals from all life stages, were washed with 1xM9 buffer thrice. The worm pellet was collected and stored at -80°C for RNA isolation. Using the Qiagen RNeasy Mini kit (no. 74104; Qiagen) RNA was isolated from the thawed pellet. The isolated RNA was treated with DNase I (Ambion's DNA-free kit AM1906) to get rid of any genomic DNA contamination. Around 3–4 µg of this RNA was then reverse transcribed into cDNA using Superscript III Reverse Transcriptase (18080093). Wild-type cDNA was also isolated and processed as a control. The sequencing of the mutated region of this cDNA was outsourced to ATGC sequencing facility at Regional Centre for Biotechnology, Faridabad, India. The sequencing confirmed that *ok2328* deletion mutation is a frameshift variant as the 16 bp insert (CCTATACTTACCTCAG) is incorporated in cDNA in the place of deleted segment, which introduces a premature stop codon after 635th amino-acid.

## Immobilization of worms for imaging experiments

The fourth larval staged worms (L4) expressing fluorescent reporters were immobilized and mounted on 5% agarose pads using 10 mM Levamisole (Sigma-Aldrich; L3008) for widefield fluorescence imaging. For confocal scanning worms were mounted on 7.5% agarose pads. We immobilized the worms for EBP-2::GFP reporter live imaging using 0.1-µm polystyrene bead suspension (Polysciences; 00876–15) on 15% agarose pads. The agarose was dissolved in 1XM9 buffer as described in previously used protocols [29].

## Widefield fluorescence imaging of mechanosensory neurons for quantifying developmental defects

The ALM and PLM neurons were visualized using a fluorescent reporter expressed under a mechanosensory neuron-specific reporter *pmec-7::*GFP *(muIs32)* using a Leica DM5000 B fluorescent microscope with a 40X air objective (NA 0.75). The L4 staged animals were mounted on 5% agarose beds and immobilized in 10mM Levamisole (Sigma-Aldrich; L3008) in 1XM9 buffer. The morphology of ALM neurons in *klp-7(0)* and suppressor mutants was qualitatively scored based on their microscopic appearance. The posterior neurite of the ALM neuron was considered a mild ectopic extension if it ended before the vulva and a strong ectopic extension if it crossed the vulva. This method was used to calculate the percentage of neurites showing the ectopic extension phenotype.

## Image acquisition and analysis of neurite length of mechanosensory neurons using a point-scanning confocal microscope

The mechanosensory neurons were visualized by expressing a fluorescent protein under a touch neuron-specific promoter either *pmec-7::*GFP (*muIs32*) or *pmec-4*::RFP (*tbIs222*). All the phenotyping or imaging experiments were performed with L4 stage animals using a Nikon confocal microscope (A1HD25) with a 60X oil objective (NA 1.4). The 1.5% of a 488nm laser was used to image GFP-tagged animals and 5% of a 561nm laser was used to image RFP-tagged animals along with a simultaneous differential interference contrast image.

To quantify the absolute and relative lengths of the anterior and posterior processes of PLM we used ImageJ software (Figs 2D, 4B, S4B–S4E and S5E). It was used to draw segmented traces along the length of the process in fluorescent channel ImageJ and then measure the length of the trace/process. The measured length of the anterior process of PLM

was normalized by the length of the trace drawn between the anal opening and the vulva of the corresponding animal, measured from the differential interference contrast image. To normalize the length of the posterior process of PLM its value was divided by the length of a trace drawn from the anal opening to the tip of the tail.

### 3-D protein modeling

The protein structure predictions were modeled using the Alphafold2 [112,113,114] on the Galaxy platform (https://usegalaxy.eu/) [108]. The molecular graphics and analysis were performed using Chimera [115] (S1C, S1D, S2E and S2F Figs).

### Molecular cloning and the generation of new transgene

The pNBR77 *pCR8::citk-1* and pNBR82 *pCR8::aspm-1* constructs were generated by cloning PCR amplified full length cDNAs in linearized pCR8 vectors by Infusion cloning (Takara, 638947). The *citk-1* cDNA was amplified using the primers AGR733 (5' CCGAATTCGCCCTTATGAACGAATCAATATATACAACGC 3') and AGR734 (5' GTCGAATTCG-CCCTTTTAGTTTTTTGGATCTTTTCAATGT 3') from yk2046g11. The *aspm-1* cDNA was amplified using the primers AGR1218 (5' TCCGAATTCGCCCTTATGGATAATAACGTTGAGG 3') and AGR1219 (5' GTCGAATTCGCCCTTTTAGAG-CATATCTTCGCATATG 3') from yk3075k20. The pCR8 cloning vector was linearized using the primers AGR607 (5' AAG-GGCGAATTCGAC 3') and AGR608 (5' AAGGGCGAATTCGAC 3').

 The pNBR77 *pCR8::citk-1* and pNBR82 *pCR8::aspm-1* constructs were then recombined with appropriate destination gateway vectors (*pmec-4::*gay and *pmec-4::gfp::gwy*) by single site LR recombination using an LR clonase enzyme (Invitrogen;11791–020) to construct pNBRGWY160 (*pmec-4::citk-1),* pNBRGWY199 (*pmec-4::gfp::citk-1)* pNBRGWY190 (*pmec-4::aspm-1)* and pNBRGWY193 (*pmec-4::gfp::aspm-1).* The clones were then injected at 1–5ng/µl to generate transgenic animals.

 *pmec-4::ebp-2::mCherry* is a gift from Dr Yishi Jin lab.

 *pttx-3*::RFP or *pmyo-2::mCherry* was used as a coinjection marker at concentrations of 60 ng/µl or 2.5ng/µl, respectively, to generate transgenic strains. The injection mixture was brought to a total concentration of 110–120 ng/µl by adding pBluescript (pBSK) plasmid to the injection mixture.

### Blast analysis

The reciprocal blast analysis was conducted for the *citk-1* protein sequence using the online NCBI protein blast platform [116,117]. The algorithm parameters were set at their default settings. The taxon-specific blast results were conducted for reciprocal blast analysis [116–118] (S2C Fig).

### Foldseek analysis

The Foldseek structural analysis was performed at default parameters in 3Di/AA mode using the online Foldseek search server [119]. The taxonomic filter was used for the taxon-specific blast results [120].

### Protein sequence alignment

The M-coffee server (https://tcoffee.crg.eu/apps/tcoffee/do:mcoffee) was used to align the protein sequences obtained from UniProt server [64,121]. SIAS was used to compare sequence similarity and sequence identity (http://imed.med.ucm.es/Tools/sias.html). Results of alignment can be seen in S1 Appendix.

### RNAi experiments

As the double mutant of *citk-1(ok2328)* and *citk-2(ok2885)* could not be generated a neuron-specific knockdown of *citk-2* in *citk-1* mutant was performed using a well-established RNAi by feeding method in worms [70–74,122] (Figs 3, 4, S3 and

S4). We used an RNAi-sensitive strain *lin-15b(n744)X; tbIs222(pmec-4::mCherry); shrSi2[pmec-3::sid-1, Cbunc-119(+)] II,* with a neuron-specific expression of a transmembrane dsRNA transporter *sid-1* under the touch neuron-specific promoter (*pmec-3*) [74] and a gentle touch neuron-specific fluorescent reporter *pmec-4::RFP (tbIs222)*. The worms were fed on HT115 *E. coli* bacteria expressing dsRNA for knockdown of gene of interest. As a positive control, the knockdown of *dhc-1* and *ama-1* was tested in this strain. The RNAi of *dhc-1*(dynein heavy chain-1) successfully produced dead embryos [123] and RNAi of *ama-1*(encodes the large subunit of RNA polymerase-II) resulted in L1 (larval stage-1) arrest phenotype [124] with a 100% penetrance. The negative control for RNAi was the HT115 bacterial strain with empty plasmid L4440. The positive control for neuronal RNAi was *unc-22* (regulates muscle contraction and relaxation) [125]–[127], and for gentle touch neuron-specific RNAi was *mec-4* (allows ligand-gated Na channel activity and is required for gentle touch sensation) [128]–[130]*,* which successfully resulted in twitching (100% penetrance) and loss of gentle touch sensation phenotype (60% penetrance), respectively. The bacterial strains carrying the dsRNA used in this work are a part of Ahringer's library [72], which was purchased from Source Bioscience (3318_Cel_RNAi_complete). The bacterial culture and feeding were performed as described in a recent paper from our lab [74]. Each dsRNA bacterial strain was grown in 4.5 ml of LB nutrient broth with Carbenicillin (50mg/ml) and Tetracycline (12.5mg/ml) antibiotics to an OD of 0.8-0.9. Once this OD was achieved, bacterial culture was centrifuged and the pellet was resuspended in 1 ml of 1XM9 containing 50mg/ml Carbenicillin, 12.5mg/ml Tetracycline, and 1.5mM IPTG. A 250µl of this bacterial suspension was used for seeding NGM plates (containing 50mg/ml Carbenicillin, 12.5mg/ml Tetracycline, and 1.5mM IPTG), followed by a 36 hrs. incubation at 25°C. 5–6 L3s were transferred to these plates, and their progenies were imaged.

## Live imaging and analysis of EBP-2::GFP dynamics

We used Nikon Ti2 widefield fluorescence microscope equipped with an Andor Zyla VSC-02284 Camera with a 560MHz readout rate for time lapse image acquisition. We captured the images at a 400ms exposure rate, a gain of 4 and 2x2 binning in a frame of 1280 × 1060 pixels using a Plan Apo VC 100 × Oil/1.4 NA objective at a frame rate of 2.44 frames per second for 2 minutes' duration.

The EBP-2::GFP imaging for RNAi experiments in Fig 3 was conducted using ZEN2 Blue software on a Zeiss Observer Z1 microscope equipped with a Yokogawa CSU-XA1 spinning disk confocal head and a Photometric Evolve electron-multiplying charge-coupled device camera for fast time-lapse image acquisition. We captured the images in a frame of 120 × 120 µm² (512 × 512 pixels2) using a 100 × /1.46 NA objective at a frame rate of 2.64 frames per second for 2 minutes' duration. We used 8.75mW of a 488-nm excitation laser to achieve the best signal-to-noise ratio for EBP-2::GFP.

To analyze the captured time-lapse images, we generated kymographs using the Plugins/ KymoReslice-Wide/Max Intensity Projection tool (https://github.com/UU-cellbiology/KymoResliceWide) [131],[132] (https://imagej.nih.gov/ij/). The kymographs were generated from two 30-µm ROIs one from the PLM anterior (from the 30u distal end to the cell body) and the other from the posterior process (from the cell body to the distal end at 30u). Kymographs are distance v/s time plots with distance along the X-axis representing axon length in micrometers and time in seconds along the Y-axis. The movement of an EBP-2::GFP comet along the axon length in a kymograph is observed as a diagonal line. We annotated the diagonal lines moving away from the cell body as "plus-end-out" microtubules (+), and the lines moving towards the cell body were annotated as "minus-end-out" microtubules (-). The fraction polarity was calculated as the relative number of plus-end-out or minus-end-out growing microtubules to the total number of microtubules. The growth length and growth duration were calculated from the slope of these tracks by using the analysis tool of ImageJ.

## Imaging and analysis of PTRN-1::GFP puncta

Using the 488nm laser of the point scanning Nikon Confocal AXR microscope, we acquired the Z-stacks of PLM neurons expressing *pmec-4*::GFP::PTRN-1 from both wild-type and mutant animals. Images were acquired using 100X 1.45 NA oil objective. Animals were immobilized using 10mM Levamisole and mounted on 7.5% agarose. The maximum intensity

projection image was background-subtracted and used to obtain the line profile of *pmec-4*::GFP::PTRN-1 puncta using the ImageJ line-profile tool. The peaks in 30µm of the anterior and posterior processes were identified from line profile using the BAR plugin of ImageJ. We counted all peaks above a minimum peak amplitude of 1 for an unbiased count of PTRN-1::GFP puncta.

## Actin imaging and kymograph analysis

Time-lapse imaging of GFP::UtCH was performed using an Olympus IX83 microscope equipped with a Yokogawa CSU-W1 spinning-disk confocal module and a Prime BSI back-illuminated sCMOS camera. Excitation was provided by a 488 nm solid-state laser operated at 5% power. Images were acquired using a 100×/1.4 NA oil objective. This optical configuration, including the SoRa module and 2×2 camera binning during acquisition, resulted in an effective pixel size of 0.13 µm per pixel.

A region of the proximal axon of the PLM (posterior lateral mechanosensory) neuron, anterior to the axonal branch point, was imaged. Time-lapse movies were acquired for 180 s at a frame rate of 1 frame per second with an exposure time of 300 ms.

Kymographs were generated from time-lapse movies using the Multi Kymograph plugin in FIJI/ImageJ. As in [79], actin polymerization events were identified and quantified as the number of events per 100 µm of axon length per minute. In kymographs, polymerization events were defined as diagonal trajectories indicative of movement, either anterograde or retrograde, that sustained across multiple frames. Events were included for analysis only if the region that polymerized earlier persisted at later time points, resulting in a triangular region of uniform intensity over the duration of the event. Apparent diagonal lines arising from the juxtaposition of stationary (vertical) regions were not counted. Transient diagonal streaks or brief intensity fluctuations giving the appearance of diagonal trajectories were excluded. Only events with uniform fluorescence intensity throughout their duration were scored, reflecting filament elongation rather than cargo movement. Each polymerization event was counted once, at initiation.

## Imaging and Analysis of GFP::CITK-1, GFP::ASPM-1 and EBP-2::mCherry puncta

Animals were immobilized using polystyrene beads and mounted on 15% agarose. Using the 100X1.54 NA oil objective of point scanning Nikon Confocal AXR microscope, we acquired the Z-stacks of PLM neurons expressing *shrEx550*(*pmec-4::gfp::citk-1* + *pmec-4::ebp-2::mCherry*) or *shrEx552*(*pmec-4::gfp::aspm-1* + *pmec-4::ebp-2::mCherry*) from both wild-type and mutant animals. GFP::ASPM-1 was imaged using 20% of 488 nm laser and EBP-2::MCherry was imaged using 10% of a 561nm laser for the *shrEx552* allele. GFP::CITK-1 was imaged using 25% of a 488nm laser and EBP-2::mCherry was imaged using 15% of 561nm laser for the *shrEx550* allele.

An image was analyzed only if both EBP-2::mCherry and GFP::CITK-1/ GFP::ASPM-1 are in focus. A single Z-slice with PLM neurite in focus in both channels was extracted and background subtracted. The multichannel slice was then split into two channels. Using the segmented line tool a 30µm trace was drawn along the PLM anterior/posterior process. This line is added to ROI for colocalization analysis using Coloc2 tool. Using default settings of this tool hundred Costes' randomizations were performed to analyze Pearson's correlation coefficient. The multichannel plot profile was generated using the BAR plugin of ImageJ.

## Live Imaging and Analysis of GFP::CITK-1, GFP::ASPM-1 and EBP-2::mCherry puncta

Animals were mounted on 15% agarose and immobilized in polystyrene beads. Using the 100X1.54 NA oil objective of point scanning Nikon Confocal AXR microscope, we acquired time-lapse live images of PLM neurons expressing *shrEx550* (*pmec-4::gfp::citk-1* + *pmec-4::ebp-2::mCherry*) or *shrEx552*(*pmec-4::gfp::aspm-1* + *pmec-4::ebp-2::mCherry*) from both wild-type and mutant animals. *shr550* was imaged at 1024X512 resolution with a line averaging of 8X, dwell time 0.4us. GFP::CITK-1 was imaged using 25% of a 488nm laser (Gain 80 and offset = 0) and EBP-2::mCherry was imaged using 15% of

561nm laser (Gain 80 and offset=0) for the *shrEx550* allele. *shr552* was imaged at 512X512 resolution with a line averaging of 2X, dwell time 1us. GFP::ASPM-1 was imaged at using 20% of 488 nm laser (Gain 45 and offset=0) and EBP-2::MCherry was imaged using 10% of a 561nm laser (Gain 40 and offset=0) for the *shrEx552* allele.

An image was processed only if both EBP-2::mCherry and GFP::CITK-1/ GFP::ASPM-1 are in focus. Images with PLM neurite in focus in both channels were bleach corrected, background subtracted and binned for visualization (See S1 and S2 Video files).

### Statistical analysis

We plotted and analyzed data using GraphPad Prism software version 9.0.2 and 9.5.1. The details of methods and statistical tests used for comparisons are given in Fig legends. The contingency plots represent the percentage of neurites showing characterized phenotypes. The bar plots represent mean and standard error of mean (SEM) for respective samples. The $X^2$ test (Fisher's exact test) was used to compare proportions. The normality of data was assessed using Shapiro-Wilk test. For Normally distributed data Bartlett's test was performed to compare variance across samples followed by Brown Forsythe and Welch ANOVA test as samples showed significant difference between variances. For samples that did not pass the normality test, the comparison was performed using Mann Whitney test for two groups or Kruskal Wallis with a post-hoc Dunn's multiple comparison test for multiple groups. In each Fig panel, P values are provided for intergroup comparisons. The respective sample size (n) and the total number of biological replicates (N) is also mentioned in the Fig legends.

### Supporting information

**S1 Table. No. of F1s screened.**
(XLSX)

**S2 Table. Identified SNPs in suppressor mutants.**
(XLSX)

**S3 Table. Descriptive results of blast analysis of *citk-1*.**
(XLSX)

**S4 Table. Strains used in this study.**
(XLSX)

**S5 Table. Transgenes generated.**
(XLSX)

**S6 Table. Tools used for analysis of WGS data.**
(XLSX)

**S1 Appendix. The sequence alignment of worm *citk-1* and *citk-2* with mammalian CIT proteins.**
(DOCX)

**S1 Fig. Related to Fig 1: Isolated variants in mec-7 (β-tubulin) and mec-12 (α-tubulin) that suppress klp-7(0).** **(A-B)** Two-dimensional schematic illustration of the exon and intron of the *mec-7* and *mec-12* gene and protein structures with positions of identified candidate mutations labeled. The scale bar is in bp (base pairs). *mec-7* and *mec-12* gene. The *mec-7(shr20)* and *mec-7(shr15)* variants are in the nucleotide binding domain of the MEC-7 protein. The *shr25* is a splice site acceptor variant at the junction of intron-2 and exon-3 which may influence RNA splicing and in effect MEC-7 protein folding or protein synthesis. The *mec-12(shr9), mec-12(shr19), mec-12(shr10)*

and *mec-12(shr1)* variants are in the GTPase domain of protein and the *mec-12(shr17), and mec-12(shr21)* variants are in the C-terminal 2-layer sandwich domain (B). **(C-D)** Alphafold [112,113,114] generated three-dimensional schematics of a MEC-7 and MEC-12 tubulin dimer with locations of *klp-7(0)* suppressor variants highlighted in magenta using Chimera [115] and rotated to display front and back views of the dimer. As depicted in Fig C and D, the *mec-7(shr15)* mutation introduces a premature stop codon, a truncated protein if formed will lack the entire α-tubulin binding domain. The non-conservative mutations in the nucleotide binding pocket of protein like *mec-7*(G148R), *mec-12*(G148S) and *mec-12*(G143E) replace a simple flexible glycine amino acid with complex amino acids with positively (arginine) or negatively (glutamic acid) charged side chains or polar side chains with hydroxyl group (serine) might lead to steric hindrance and hence nucleotide binding and protein folding. The *mec-12* mutation P32S is in the P loop and the G321R, S277L are present on the inter-dimer surface. These substitutions likely impact microtubule polymerization, nucleotide binding, or heterodimer interactions. The *mec-12*(C129Y) mutation replaces a small cysteine nucleotide capable of forming disulfide bonds with a large polar and aromatic tyrosine in interdimer surface. The mutation could disrupt the protein structure or stability or could even influence interdimer interactions. These effects on protein could eventually lead to changes in the fifteen protofilament architecture of microtubules in the mechanosensory neurons.
(TIFF)

**S2 Fig. Related to Fig 2: Mapping and characterization of *shr11* variant. (A)** Normalized variant density plot of Chromosome II obtained by EMS density mapping analysis of *shr11* variant from the whole genome sequencing data using the method described in [110]. The variant density is normalized as the number of EMS induced variants per 500Mb (gray bars) along the chromosomal length in Mb. It peaks at the genomic position linked to *shr11* variant. **(B)** Quantification of the percentage of ALM exhibiting ectopic extension phenotype in the wild-type, *klp-7(0), klp-7(0); citk-1(shr11), klp-7(0); citk-1(shr11); citk-1*(genomic DNA)[+] backgrounds, where *citk-1* (genomic DNA) is the fosmid WRM062dD08. N = 3–5 independent replicates, n (number of neurons) = 45–60. P values from 2x2 Fischer's exact test. **(C-D)** Quantification of the normalized length of PLM posterior process in **(C)** *klp-7(0), citk-1(shr11); klp-7(0), citk-1(ok2328); klp-7(0), citk-1(shr11); klp-7(0); citk-1* genomic DNA [+] and *citk-1(shr11); klp-7(0); citk-1* cDNA [+] where genomic *citk-1* is the *citk-1* fosmid WRM062dD08 and the *citk-1(cDNA)* is expressed under mechanosensory neuron specific promoter *pmec-4* and in **(D)** *klp-7(0)* and *citk-2(ok2885); klp-7(0)* backgrounds. Normalized length of PLM posterior = (Absolute length/distance between the PLM cell body to the tip of the tail for the posterior neurite). For (D) N = 3–4 independent replicates, n (number of neurons) = 11–50. P values from Kruskal Wallis test followed by Dunn's multiple comparisons. For (E) N = 3–4 independent replicates, n (number of neurons) = 42–45. **(E)** Graphical overview of the distribution of top blast hits [117] on the query sequence, W02B8.2/CITK-1, represented by the pink bar. The hits are shown aligned to the regions of query, below in color coded bars. **(F)** Heat map showing the percentage similarity and percentage identity shared between citron rho interacting kinase proteins from different species. **(G)** The Alphafold predicted structures of CITK-1, CITK-2, and the kinase-less isoform of mammalian CIT (CIT-N). The inset shows the pleckstrin-homology domain (blue) and the citron-homology domain (magenta). The matrix on the right shows the expected position error in the predicted structures as given by Alphafold. **(H)** The Foldseek [119] predicted structural alignments and Tm alignment scores between CITK-1 and CITK-2, CITK-1 and CIT-N, and CITK-2 and CIT-N. Tm (Template modelling score) is a quantitative measure of protein structure similarity and lies between the values 0 and 1, where 1 indicates a perfect match. A Tm score below 0.3 indicates a random structural similarity [133]. **(I-J)** The schematic of the exons and introns of the (I) *citk-1* and (J) *citk-2* gene and the genomic positions of (I) *shr11, ok2328* and (J) *ok2885* alleles in the gene locus. Below the gene structure is the schematic of (I) CITK 1 and (J) CITK-2 proteins and the deleted regions in the (I) *ok2328* and (J) *ok2885* alleles. **(K)** Quantification of absolute length of ALM posterior process in wild-type control and in wild-type animals with transgenic overexpression of *shrEx463(Pmec-4::citk-1).* PLM neurons were visualized using *muIs32(Pmec-7::GFP)* transgene. N = 2 independent replicates, n (number of neurons) = 25–50. P values from student's

T-test. For D-E, K, Error bars represent SEM (Standard error mean), ***, P<0.001; **, P<0.01, *, P<0.05, ns, not significant.
(TIFF)

**S3 Fig. Related to Fig 3: Microtubules are more dynamic in the PLM anterior process of *citk* mutant animals. (A-B)** Histograms depicting the growth length **(A)** and growth duration **(B)** quantified by determining the net pixel shift in the X and Y axis of kymographs obtained from PLM anterior processes of same backgrounds as Fig 3. N = 3–5 independent replicates, n (number of tracks) = 120–1081. **(C)** Quantification of fraction of plus-end-out microtubules (number of plus-end-out MTs/Total number of MTs) in the anterior process of given backgrounds. N = 3–5 independent replicates, n (number of tracks) = 6–18 **(D)** Quantification of number of growing microtubules per kymograph in the posterior process of PLM neurons for A-D, N = 3 independent replicates, n (number of neurons) = 6–26. **(E)** Representative confocal images of PLM neurons in L4 staged wild-type and *citk-1(0)* animals expressing transgene *juEx6455(pmec-4::gfp::ptrn-1)*. Black arrowheads point to GFP::PTRN-1 puncta. The blue and orange dotted rectangle represent the 30μm ROI in the anterior and posterior process of PLM neuron. **(F)** Quantification of the number of GFP-PTRN-1 puncta seen in the 30μm ROIs in the anterior and posterior process of PLM neurons (as shown in Fig 3A and 3B). **(G-H)** Representative kymographs obtained by time lapse live imaging of PLM neurons expressing actin reporter, GFP::UtCH (G) and quantification of the actin trail density/100um/min in distal region of PLM neurons in WT and *citk-1(0)* animals (A-D, F, H) ***, P<0.001; **, P<0.01; *, P<0.05, (A-D) P values from Kruskal Wallis test followed by Dunn's multiple comparison test. The groups separated by dotted lines were analyzed independent of each other. (F) P values from ANOVA with Tukey's multiple comparison test. (H) P values from Mann Whitney U Test. (A-D, F) Error bars represent SEM (Standard error mean), ns, not significant.
(TIFF)

**S4 Fig. Related to Fig 4: *citk-1* single mutants display a short PLM posterior process. (A-A')** Representative confocal images and schematics of the ALM and PLM mechanosensory neurons in L4 stage wild-type and *citk-1(ok2328)* mutant animals. These neurons are labeled by *muIs32(pmec-7::GFP)* reporter. A dotted blue arrow marks the distal tip of PLM anterior process. The white arrow points to the ventral synaptic branch. White double-headed arrow is drawn along the length of PLM posterior process. Vulva on the ventral side of animal is marked by an asterisk (*), and anus is marked by a generic symbol (¤). **(B-C)** Quantification of the normalized length of PLM posterior process (B) and anterior process (C) in the wild-type, *citk-1(shr11), citk-1(ok2328)* and *citk-2(ok2885)* animals. (The PLM posterior process was also rescued by *shrEx514(Pcitk-1::citk-1),* extrachromosomal expression of a fosmid, WRM062dD08, containing complete genomic locus of *citk-1,* in *citk-1(ok2328)* mutant animals as quantified in (B). For B-C, N = 3–5 independent replicates, n (number of neurons) = 30–48. **(D)** Quantification of percentage of PLM neurons displaying a defect in the positioning or count of ventral synaptic branch in the wild-type, *citk-1(shr11), citk-1(ok2328)* and *citk-2(ok2885)* mutant animals. N = 3–5 independent replicates, n (number of neurons) = 51–68. **(E)** Quantification of the normalized length of PLM anterior process in the wild-type control and *citk-1(0)* RNAi sensitive background animals fed on bacteria expressing either empty vector (control) or *citk-1* or *citk-2* ds RNA for RNAi mediated knockdown. N = 3–5 independent replicates, n (number of neurons) = 17–34. **(F)** Quantification of number of short ectopic protrusions/ PLM neuron anterior process in the wild-type control and *citk-1(0)* RNAi sensitive background animals fed on bacteria expressing either empty vector (control) or *citk-1* or *citk-2* ds RNA for RNAi mediated knockdown. N = 3–5 independent replicates, n (number of neurons) = 32–56. Error bars represent SEM (Standard error mean), ns, not significant; ***, P<0.001; **, P<0.01; *, P<0.05. For B-C and E-F, P values from Kruskal Wallis test followed by Dunn's multiple comparison. For D, P values from 2x2 Fisher's exact test.
(TIF)

**S5 Fig. Related to Fig 5: ASPM-1 works in the same genetic pathway as CITK-1 and CITK-2. (A-B)** Histograms depicting the growth length (A) and growth duration (B) of the EBP-2::GFP bound microtubules in the anterior process of

PLM neurons in the wild-type and *aspm-1(0)* mutants, quantified by determining the net pixel shift in the X and Y axis (of kymographs) respectively. **(C)** Plot of fraction of plus-end out microtubules in PLM anterior process of respective genotypes. The wild-type, *aspm-1(ts), aspm-1(ts); shrEx546(+), citk-1(0)* and *aspm-1(ts); citk-1(0)* animals were grown at non-permissive temperature of 25°C. *shrEx546* transgene expresses *pmec-4::aspm-1*. For A-C, N = 3–4 independent replicates, for A-B, n (number of tracks) = 112–756. P values from Kruskal Wallis test followed by Dunn's multiple comparison and for C, n (number of PLM neurons) = 13–29. **(D-G)** Quantification of microtubule growth parameters (D-F) and fraction polarity (G) in PLM neurons of wild-type, *aspm-1(ts), citk-2(ok2885)* and *aspm-1(ts); citk-2(ok2885)* mutant backgrounds reared at non-permissive temperature of 25°C. For D-G, N = 3–4 independent replicates, for D, G, n (number of PLM neurons) = 12–23 and for E-F, n (number of tracks) = 512–1665. For D-G, P values from Kruskal Wallis test followed by Dunn's multiple comparison. **(H-J)** Quantification of microtubule growth length(H), growth duration(I), and fraction polarity(J) in the anterior and posterior processes of WT and WT animals overexpressing *aspm-1* as transgene *shrEx530(pmec-4::aspm-1)(++).* For H-J, N = 3 biological replicates, for H-I, n(no. of tracks) = 349–871 and for J, n(number of neurons) = 12–18. All animals in A-B and D-I express transgene *juIs338(pmec-4::EBP-2::GFP).* For H-J, P values from unpaired Student's T-test. Error bars represent SEM (Standard error mean), ns, not significant; ***, P < 0.001; **, P < 0.01; *, P < 0.05.
(TIFF)

**S6 Fig. Related to Fig 5: ASPM-1 is required in post-mitotic PLM neurons to regulate microtubule dynamics. (A)** Schematic illustrating control and test growth condition-1 and 2 animals were subjected-to for experiments in B-G. Animal growth condition-1 in pink illustrated rearing of animals at 25°C. Animal growth condition-2 in sky blue illustrates animals reared at 20°C. L1 progeny of these animals is transferred and grown to L4 stage at 25°C for experiments. **(B-G)** Quantification of microtubule growth parameters in PLM neurons of WT and *aspm-1(ts),* in control or test conditions. Kymographs were obtained from same ROI as described in Fig 3. The color of the background panel denotes the growth condition for respective datasets. For B-G, N = 3–4 biological replicates, For B, E-G, n(number of neurons) = 12–24, for C-D, n(number of tracks) = 805–1667. All animals in B-G express transgene *juIs338(pmec-4::EBP-2::GFP).* For B-G, Error bars represent SEM (Standard error mean), ns, not significant; ***, P < 0.001; **, P < 0.01; *, P < 0.05, P values from Kruskal Wallis test followed by Dunn's multiple comparison.
(TIFF)

**S7 Fig. Related to Fig 5: ASPM-1 mutation can suppress *klp-7(0)* ectopic extension phenotype. (A-B)** Representative confocal images of ALM and PLM neurons in L4 staged *klp-7(0)* and *aspm-1(0); klp-7(0)* mutant animals grown at non-permissive temperature. The neurons are labeled by *muIs32(pmec-7::GFP)* reporter. However, the image has been inverted in ImageJ for presentation purposes. The ALM extends two mild posterior extensions (pink arrowhead) in *klp-7(0)* animal. The *aspm-1(0); klp-7(0)* double mutant shows no extension. The blue dotted arrow marks the distal tip of anterior process of PLM neuron, and the black double-headed arrow is drawn along the length of the PLM posterior process. Black solid arrows point to ventrally extended synaptic branches. **(B)** Quantification of the length of ALM posterior process in *klp-7(0), aspm-1(ts)* and *aspm-1(ts); klp-7(0)* mutants. N = 3 and n (number of neurons) = 21–46. **(C)** Representative confocal image showing distribution of PTRN-1::GFP puncta in the anterior and posterior process of PLM neurons in WT and *aspm-1(ts)* animals expressing transgene *juEx6455(pmec-4::gfp::ptrn-1).* The animals were reared at 25°C. The blue and orange dotted rectangle represent the 30μm ROI in the anterior and posterior process of PLM neuron. Black arrowheads point to GFP::PTRN-1 puncta. **(D)** Quantification of number of GFP::PTRN-1 puncta in WT and *aspm-1(ts)* grown at 20°C (in sky blue) and 25°C(in pink). The number of GFP::PTRN-1 puncta were quantified from 30μm ROI. N = 3 and n (number of neurons) = 15–23. For B, D Error bars represent SEM (Standard error mean). ***, P < 0.001; **, P < 0.01; * P < 0.05 and ns, not significant. For B, P values from Kruskal Wallis test followed by Dunn's multiple comparison. For D, P values from ANOVA with Tukey's multiple comparison test.
(TIFF)

**S1 Video. Related to Fig 6: Live Imaging of GFP::CITK-1 and EBP-2::mCherry.** The representative time-lapse video of GFP::CITK-1(in green) and EBP-2::MCherry(in magenta) tracks in anterior processes of PLM neurons in WT background. EBP-2::mCherry associates with the growing plus-ends of microtubules. White arrows track the EBP-2::mCherry colocalized GFP::CITK-1 comets (in white), moving away from the cell body. Videos were captured using a Nikon AXR point-scan confocal microscope at a speed of 1.48 frames per second and played at 7 frames per second.
(MP4)

**S2 Video. Related to Fig 6: Live Imaging of GFP::ASPM-1 and EBP-2::mCherry.** The representative time-lapse video of GFP::ASPM-1(in green) and EBP-2::MCherry(in magenta) tracks in anterior processes of PLM neurons in WT background. White arrows track the EBP-2::mCherry(MT plus ends) colocalized GFP::ASPM-1 comets (in white), moving away from the cell body. Videos were captured using a Nikon AXR point-scan confocal microscope at a speed of 1 frame per second and played at 7 frames per second.
(MP4)

## Acknowledgments

We are grateful to Yuji Kohara for sharing cDNAs. We thank the National BioResource Project (NBRP), Japan, and the Caenorhabditis Genetics Center (CGC) for providing strains. NBRP is supported by the Japanese government, and CGC is supported by the NIH Office of Research Infrastructure Programs (P40 OD010440). We acknowledge the contributions of Pankajam Thyagarajan, Dharmendra Puri, Sanskriti Swamy, Kapil Raj, Sreyashi Chandra, Akanksha Goyal, and Debopriya Roy in initiating the EMS mutagenesis. We also thank Deepanshu Aswal for his diligent assistance with PCR genotyping and *ebp-2* imaging. We further thank Chakshu Mangal for her expert technical support in PTRN-1::GFP image collection. For all the genotyping and cloning-related sequencing, we thank the sequencing facility at Genomics Facility of the Advanced Technology Platform Centre (ATPC), which is managed by the Regional Centre for Biotechnology (RCB) and is funded by the Department of Biotechnology (Grant No. BT.MED-II/ATPC/BSC/01/2010).

## Author contributions

**Conceptualization:** Sunanda Sharma, Anindya Ghosh-Roy.

**Data curation:** Sunanda Sharma.

**Formal analysis:** Sunanda Sharma, Keerthana Ponniah, Ishanee Bandyopadhyay, Devyani Vadawale.

**Funding acquisition:** Sandhya P Koushika, Anindya Ghosh-Roy.

**Investigation:** Sunanda Sharma, Keerthana Ponniah.

**Methodology:** Sunanda Sharma, Keerthana Ponniah, Sandhya P Koushika.

**Project administration:** Anindya Ghosh-Roy.

**Resources:** Anindya Ghosh-Roy.

**Supervision:** Sandhya P Koushika, Anindya Ghosh-Roy.

**Validation:** Sunanda Sharma, Keerthana Ponniah.

**Visualization:** Sunanda Sharma, Keerthana Ponniah, Ishanee Bandyopadhyay, Devyani Vadawale.

**Writing – original draft:** Sunanda Sharma, Anindya Ghosh-Roy.

**Writing – review & editing:** Sunanda Sharma, Anindya Ghosh-Roy.

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
