## [Decision Letter · Decision Letter 0]

13 Jun 2025

PGENETICS-D-25-00512

The cytokinesis associated proteins CITK and ASPM-1 regulate neuronal microtubule dynamics and polarity in C. elegans.

PLOS Genetics

Dear Dr. Ghosh-Roy,

Thank you for submitting your manuscript to PLOS Genetics. After careful consideration, we feel that it has merit but does not fully meet PLOS Genetics's publication criteria as it currently stands. In particular, reviewers raised questions about whether ASPM-1 and CITK-1/2 function in the same pathway (suggesting genetic tests to examine this); whether the worm proteins examined were true paralogs of the human proteins; and had concerns with the clarity of the writing. We invite you to submit a revised version of the manuscript that addresses the points raised during the review process.

Please submit your revised manuscript within 60 days Aug 12 2025 11:59PM. If you will need more time than this to complete your revisions, please reply to this message or contact the journal office at plosgenetics@plos.org. Please include the following items when submitting your revised manuscript:

We look forward to receiving your revised manuscript.

Kind regards,

Jeremy Nance

Academic Editor

PLOS Genetics

Pablo Wappner

Section Editor

PLOS Genetics

Aimée Dudley

Editor-in-Chief

PLOS Genetics

Anne Goriely

Editor-in-Chief

PLOS Genetics

**Journal Requirements:**

At this stage, the following Authors/Authors require contributions: Sunanda Sharma, Keerthana Ponniah, and Anindya Ghosh-Roy. Please ensure that the full contributions of each author are acknowledged in the "Add/Edit/Remove Authors" section of our submission form.

The list of CRediT author contributions may be found here: https://journals.plos.org/plosgenetics/s/authorship#loc-author-contributions

https://journals.plos.org/plosgenetics/s/submission-guidelines#loc-parts-of-a-submission

5) Please ensure that the funders and grant numbers match between the Financial Disclosure field and the Funding Information tab in your submission form. Note that the funders must be provided in the same order in both places as well. State what role the funders took in the study. If the funders had no role in your study, please state: "The funders had no role in study design, data collection and analysis, decision to publish, or preparation of the manuscript.".

**Reviewers' comments:**

Reviewer's Responses to Questions

**Comments to the Authors:**

Reviewer #1: Review is uploaded as an attachment

Reviewer #2: The authors use an elegant forward genetic screen to identify modulators of neurite outgrowth linked to microtubule growth in neurons using the advantages of C. elegans. Using a suppressor screen, they identify citk as a regulator of neurite outgrowth through modulation of microtubule growth dynamics/stability. This is a unique role for this protein previously studied primarily as a regulator of cell division previously. The role of citk in neurite extension is further supported by the fact that ASPM, a protein that functions with CITK during cell division, causes similar defects in neurons when mutated. Together, this data argues for roles for these cell division-associated proteins in post-mitotic neurons. Despite my original enthusiasm for this work, there are some missing pieces that are required to provide mechanistic insight into how CITK functions to regulate microtubules and neurite outgrowth. Furthermore, the homology between W02B8.2 and CITK should be confirmed by rescue analysis with a mammalian ortholog. Finally, on a more stylistic point, the writing is drawn out and needs to be more pointed. Long sentences are elegant but, in some cases, the meaning of the sentence is lost. I would encourage the authors to use more succinct language to convey the specific point being made. This is particularly true in the discussion which needs substantial revision. These and other points are discussed below.

1) Section “PLM mechanosensory neurons displayed growth and guidance defects on loss of both citk1 and citk2” can be significantly shortened. Specifically, the discussion of the partially penetrant guidance/positioning phenotypes could be severely abbreviated or eliminated as no follow up mechanistic work was done. It distracts the reader from the main points of the paper if not better integrated.

2) Confirmation that W02B8.2 is the worm CITK homolog requires validation. Can the phenotype be suppressed by expression of the mammalian version of CITK or CITN? Both?

3) CIT-N (variant of CITK) has been implicated in Golgi positioning in hippocampal neurons - a dynein-dependent phenotype. Dynein has also been linked to microtubule polarity due to its role in microtubule sliding. Can you expound on the citk loss of function and include a discussion on how the phenotypes could be similar/different to the work on CIT-N in hippocampal neurons? Do dynein hypo- or hyper-morphs give you a similar defect in neuritis length and microtubule polarity? This point should be addressed either with further discussion or experiments as this phenotype has been clearly shown in post-mitotic neurons previously.

4) As you mention, CIT-K regulates actin dynamics in neurons. I think it is important to discern whether the effect you see is due to microtubules directly or through actin regulation. It would be useful to look at actin dynamics with live imaging or even actin structure to see if there is any reason to think the microtubule defects you see are secondary to actin disruption. Additionally, a more detailed assessment of the direct nature of the microtubule modifications is necessary in the discussion. This would be aided by a detailed assessment of the cellular localization of citk in these neurons. Does it localize to microtubule plus ends in neurites?

5) ASPM mutation gives the same phenotype as CITK but it is not delineated if this is because it is in parallel or the same cellular pathway. Is the CITK binding domain in ASPM known (or vice versa)? Could this be used in rescue analysis to further determine if these two are functioning together in neurite outgrowth? ASPM localizes to microtubule plus ends and recruits CITK during cell division. Is plus end localization of CITK lost in ASPM mutants or important for CITK function? Alternatively, do partial or full loss of ASPM and CITK together produce an additive phenotype?

6) The writing could be made a bit more straightforward. The long/wandering nature of the text makes it a bit difficult to follow. This is true in the discussion most clearly. For example, the paragraph that starts at line 555 is discussion the axon phenotype similarities to previous reports. Then branches into how citk may regulate actin for 1 sentence then astral microtubules in the next without linking these ideas at all. The following paragraph is 2 sentences, with the first one stretching 5 lines. It is unclear what is gained from this paragraph.

7) The discussion is a list of things that “should be tested” but weren’t in this study. Some things, particularly the actin cytoskeleton, should be addressed so more definitive conclusions can be drawn. The discussion makes it sound like this is just a replication of previous work without any progress. I don’t think this is the case. A stronger discussion of what has been learned/advanced is needed.

Reviewer #3: In this manuscript, Sharma et al., studied the function of CITK proteins and ASPM-1 in regulating microtubule dynamics and polarity in C. elegans neurites. They took advantage of the genetic system and identified citk-1 mutations from a genetic screen that is designed to isolate MT regulators. Then they demonstrated that citk-1 and citk-2 functions redundantly in regulating MT dynamics and polarity in the PLM neurons. Because of previous literature linking these two proteins to ASPM1, they tested aspm1’s involvement and found that it also functions in PLM to regulate MTs. The experiments are well designed and executed. The findings are novel and interesting, but lack mechanisms. I support the publication of a revised manuscript. My main suggestions are two-fold: one is on writing and one is on the timing of the function of these proteins.

Major points.

1. The introduction does not reflect an updated set of papers regarding MT organization in neurons. I would suggest the authors to systematically cite papers and separate them into the invertebrate and vertebrate systems because the MT polarity in neurites are somewhat different and the invertebrate systems often include in vivo analyses.

2. Despite of the very interesting observations on MT dynamics and polarity phenotypes, the mechanisms of these defects are unclear. I am not suggesting that the authors need to provide a thorough mechanism before this paper can be published. However, it might be very informative to know whether the function of citk and aspm1 are mitotic or postmitotic. In other words, are all the phenotypes caused by defects in early developmental stages (like during mitosis that generates PLM) or are these genes required for the maintenance of the MT dynamics and polarity. Since aspm1 is a temperature allele, that experiment can be easily performed. Are there ways to generate conditional alleles of citk-1 or -2 to test the requirement of these genes for the maintenance of MT properties?

Minor points.

3. Line 60 “y-TURC” should be “g-TuRC”.

4. The Y-axis of figure 2D is not labeled.

5. In Figure 1, it seems strange to use mild and strong extension as a measure instead of directly measuring the length of the posterior process.

6. It will help to comment on the nature of the b- and a-tubulin mutations in the suppressors. The stop codon mutations are loss of function. Can the authors comment on why the loss of tubulin does not lead to growth defects in the anterior process.

7. Line 294, “… PLM neurons of citk mutant animals seemed to be significantly

295 higher than the number of plus-end-out comets implying a trend…”

Change the statement to a more formal one.

8. Line414. “…Figure 4E-G)…” There is no 4G in the figure.

**Have all data underlying the figures and results presented in the manuscript been provided?**

Large-scale datasets should be made available via a public repository as described in the *PLOS Genetics*
data availability policy, and numerical data that underlies graphs or summary statistics should be provided in spreadsheet form as supporting information., and numerical data that underlies graphs or summary statistics should be provided in spreadsheet form as supporting information., and numerical data that underlies graphs or summary statistics should be provided in spreadsheet form as supporting information., and numerical data that underlies graphs or summary statistics should be provided in spreadsheet form as supporting information.

Reviewer #1: Yes

Reviewer #2: None

Reviewer #3: Yes

PLOS authors have the option to publish the peer review history of their article (what does this mean?). If published, this will include your full peer review and any attached files.). If published, this will include your full peer review and any attached files.). If published, this will include your full peer review and any attached files.). If published, this will include your full peer review and any attached files.

...

Reviewer #1: No

Reviewer #2: No

Reviewer #3: No

**Figure resubmission:**
---

## [Decision Letter · Decision Letter 1]

3 Mar 2026

PGENETICS-D-25-00512R1

Worm orthologues of cytokinesis-associated proteins CIT and ASPM regulate neuronal microtubule dynamics and polarity in C. elegans

PLOS Genetics

Dear Dr. Ghosh-Roy,

Thank you for submitting your manuscript to PLOS Genetics. After careful consideration, we feel that it has merit but does not fully meet PLOS Genetics's publication criteria as it currently stands. Therefore, we invite you to submit a revised version of the manuscript that addresses the minor points raised during the review process.

Please submit your revised manuscript within by Apr 02 2026 11:59PM. If you will need more time than this to complete your revisions, please reply to this message or contact the journal office at plosgenetics@plos.org. Please include the following items when submitting your revised manuscript:

We look forward to receiving your revised manuscript.

Kind regards,

Jeremy Nance

Academic Editor

PLOS Genetics

Pablo Wappner

Section Editor

PLOS Genetics

Aimée Dudley

Editor-in-Chief

PLOS Genetics

Anne Goriely

Editor-in-Chief

PLOS Genetics

**Reviewers' comments:**

Reviewer's Responses to Questions

**Comments to the Authors:**

Reviewer #1: The authors have revised the manuscript by conducting new experiments to address some of the major issues that were brought up in the initial round of review. In particular, the genetic experiments provide support for the hypothesis that ASPM-1 and CITK-1/2 function in the same pathway; a model that was presented in the previous version but not examined experimentally. Moreover, clarification in the writing and the way figures are presented have made the manuscript easier to follow and more compelling. I do still have some minor suggested revisions outlined below, but otherwise feel that the work is acceptable for publication.

a) Line 107: “in the microtubule dynamics may lead to distinct morphological changes…”. I suggest removing the word “may”. These phenotypes are well-established, and you show examples in this paper. So, we know for sure that disruption of MT dynamics cause morphological changes.

b) Lines 132-133: “in-situ hybridization experiments, …, showed localization of CITK protein in a specific neuronal population in the cortex and thalamus …”. In situ experiments show mRNA localization, not protein. I would reword to say “showed CITK expression in a specific …”.

c) Line 143: The sentence starting “Live imaging of EBP-2::GFP reporter …”. It is not clear here that the authors are starting to describe their own data. I would start this sentence instead saying “Here, using live imaging of an EBP-2::GFP reporter …”. Or maybe put a summarizing sentence before starting to describe the data. Something like “Here we show that the C. elegans orthologs of vertebrate of the CIT kinase, also regulate MT dynamics …”. This would make it clear that your shifting from background information to summarizing what you have discovered.

d) Lines 400-402: The sentence “Notably, the fraction of minus-end-out microtubules in the posterior process increases significantly, mirroring the effects of citk-1 overexpression in wild-type animals (Fig S5J)” should be re-written so that the call to the supplemental figure is in the right location, and the authors should also call-out the data in wild-type. So, change to “Notably, the fraction of minus-end-out microtubules in the posterior process increases significantly (Fig S5J), mirroring the effects of citk-1 overexpression in wild-type animals (Fig 3D)”

e) Line 408: “suggesting an epistatic genetic relationship”. I don’t think this is the correct term to use. Epistasis implies that the mutants have different phenotypes, and that in the double mutant one phenotype “covers-up” (epistasis literally means “stands upon”) the other. This is most evident when one mutation suppresses the phenotype of the other. What you are discovering here is that combining two mutations does not make the phenotype worse, so I would just say that the results are consistent with the genes functioning in the same pathway.

f) Line 414: “equivalent to animals reared at the non-permissive temperature”. I find this confusing. if the mutation is embryonic lethal, how were the authors rearing the animals at the non-permissive temperature continuously? For the continuous non-permissive conditions, were eggs laid at the permissive temperature then shifted? They need to provide more details here, and in the Methods as to how these animals were maintained and when the temperature shifts were done in all their experiments with aspm-1(ts). The description in the methods (“grown at non-permissive temperature of 25°C according to the experimental paradigm”) is too vague.

Reviewer #2: This revision has taken into account most of my concerns. I am still a bit worried about the strength of the assertion that these genes are the kinase-less orthologs of the mammalian gene without rescue showing functional redundancy. While cloning the gene may be difficult there are many mammalian orthologs already cloned by other groups. However, at this point, this concern can be remedied by tempering the language a bit. For example, on line 212, I would suggest changing the word "confirmed" which is very strong given the information presented to "suggests" or "argues" or "supports".

Reviewer #4: The authors have addressed all of the major points raised by the reviewers, and I support publication.

**Have all data underlying the figures and results presented in the manuscript been provided?**

Large-scale datasets should be made available via a public repository as described in the *PLOS Genetics*
data availability policy, and numerical data that underlies graphs or summary statistics should be provided in spreadsheet form as supporting information., and numerical data that underlies graphs or summary statistics should be provided in spreadsheet form as supporting information., and numerical data that underlies graphs or summary statistics should be provided in spreadsheet form as supporting information., and numerical data that underlies graphs or summary statistics should be provided in spreadsheet form as supporting information.

Reviewer #1: Yes

Reviewer #2: None

Reviewer #4: Yes

PLOS authors have the option to publish the peer review history of their article (what does this mean?). If published, this will include your full peer review and any attached files.). If published, this will include your full peer review and any attached files.). If published, this will include your full peer review and any attached files.). If published, this will include your full peer review and any attached files.

...

Reviewer #1: No

Reviewer #2: No

Reviewer #4: No

**Figure resubmission:**
---

## [Editor Report · Decision Letter 2]

26 Mar 2026

Dear Dr Ghosh-Roy,

We are pleased to inform you that your manuscript entitled "Worm orthologues of cytokinesis-associated proteins CIT and ASPM regulate neuronal microtubule dynamics and polarity in C. elegans." has been editorially accepted for publication in PLOS Genetics. Congratulations!

Yours sincerely,

Jeremy Nance

Academic Editor

PLOS Genetics

Pablo Wappner

Section Editor

PLOS Genetics

Aimée Dudley

Editor-in-Chief

PLOS Genetics

Anne Goriely

Editor-in-Chief

PLOS Genetics

BlueSky: @plos.bsky.social

Comments from the reviewers (if applicable):

**Data Deposition**

If you have submitted a Research Article or Front Matter that has associated data that are not suitable for deposition in a subject-specific public repository (such as GenBank or ArrayExpress), one way to make that data available is to deposit it in the Dryad Digital Repository. As you may recall, we ask all authors to agree to make data available; this is one way to achieve that. A full list of recommended repositories can be found on our . As you may recall, we ask all authors to agree to make data available; this is one way to achieve that. A full list of recommended repositories can be found on our . As you may recall, we ask all authors to agree to make data available; this is one way to achieve that. A full list of recommended repositories can be found on our . As you may recall, we ask all authors to agree to make data available; this is one way to achieve that. A full list of recommended repositories can be found on our website....

http://datadryad.org/submit?journalID=pgenetics&manu=PGENETICS-D-25-00512R2

Additionally, please be aware that our data availability policy requires that all numerical data underlying display items are included with the submission, and you will need to provide this before we can formally accept your manuscript, if not already present. requires that all numerical data underlying display items are included with the submission, and you will need to provide this before we can formally accept your manuscript, if not already present. requires that all numerical data underlying display items are included with the submission, and you will need to provide this before we can formally accept your manuscript, if not already present. requires that all numerical data underlying display items are included with the submission, and you will need to provide this before we can formally accept your manuscript, if not already present.

**Press Queries**

If you or your institution will be preparing press materials for this manuscript, or if you need to know your paper's publication date for media purposes, please inform the journal staff as soon as possible so that your submission can be scheduled accordingly. Your manuscript will remain under a strict press embargo until the publication date and time. This means an early version of your manuscript will not be published ahead of your final version. PLOS Genetics may also choose to issue a press release for your article. If there's anything the journal should know or you'd like more information, please get in touch via plosgenetics@plos.org....

---

## [Editor Report · Acceptance letter]

PGENETICS-D-25-00512R2

Worm orthologues of cytokinesis-associated proteins CIT and ASPM regulate neuronal microtubule dynamics and polarity in C. elegans.

Dear Dr Ghosh-Roy,

We are pleased to inform you that your manuscript entitled "Worm orthologues of cytokinesis-associated proteins CIT and ASPM regulate neuronal microtubule dynamics and polarity in C. elegans." has been formally accepted for publication in PLOS Genetics! Your manuscript is now with our production department and you will be notified of the publication date in due course.

With kind regards,

Anita Estes

PLOS Genetics

On behalf of:
